# Unraveling of cocatalysts photodeposited selectively on facets of BiVO$_4$ to boost solar water splitting

Yu Qi[1,7], Jiangwei Zhang [1,7], Yuan Kong[2,7], Yue Zhao[1], Shanshan Chen [3], Deng Li[1], Wei Liu[1], Yifan Chen[4], Tengfeng Xie[4], Junyan Cui[5], Can Li [1✉], Kazunari Domen [3,6] & Fuxiang Zhang [1✉]

Bismuth vanadate (BiVO$_4$) has been widely investigated as a photocatalyst or photoanode for solar water splitting, but its activity is hindered by inefficient cocatalysts and limited understanding of the underlying mechanism. Here we demonstrate significantly enhanced water oxidation on the particulate BiVO$_4$ photocatalyst via in situ facet-selective photo-deposition of dual-cocatalysts that exist separately as metallic Ir nanoparticles and nano-composite of FeOOH and CoOOH (denoted as FeCoO$_x$), as revealed by advanced techniques. The mechanism of water oxidation promoted by the dual-cocatalysts is experimentally and theoretically unraveled, and mainly ascribed to the synergistic effect of the spatially separated dual-cocatalysts (Ir, FeCoO$_x$) on both interface charge separation and surface catalysis. Combined with the H$_2$-evolving photocatalysts, we finally construct a Z-scheme overall water splitting system using [Fe(CN)$_6$]$^{3-/4-}$ as the redox mediator, whose apparent quantum efficiency at 420 nm and solar-to-hydrogen conversion efficiency are optimized to be 12.3% and 0.6%, respectively.

[1] State Key Laboratory of Catalysis, Dalian Institute of Chemical Physics, Chinese Academy of Sciences, Dalian 116023, China. [2] Hefei National Laboratory for Physical Sciences at the Microscale and Department of Chemical Physics, University of Science and Technology of China, Hefei, Anhui 230026, China. [3] Research Initiative for Supra-Materials (RISM), Shinshu University, 4-17-1 Wakasato, Nagano 380-8553, Japan. [4] College of Chemistry, Jilin University, Changchun, Jilin 130012, China. [5] School of Material Science and Engineering, Zhengzhou University, No. 100 Science Avenue, Zhengzhou 450001, China. [6] Office of University Professors, The University of Tokyo, 7-3-1 Hongo, Bunkyo-ku, Tokyo 113-8656, Japan. [7] These authors contributed equally: Yu Qi, Jiangwei Zhang, Yuan Kong. ✉email: canli@dicp.ac.cn; fxzhang@dicp.ac.cn

Particulate photocatalytic overall water splitting (OWS) based on inorganic semiconductor materials with relative good photostability has been demonstrated as one of the most promising ways of realizing scalable and economically viable solar hydrogen production to address energy- and environment-related issues[1–12]. To achieve high solar-to-hydrogen (STH) energy conversion efficiency, it is necessary to increase the quantum efficiency of photocatalytic OWS over a wide range of wavelengths, particularly the use of visible light[11,12]. However, extended visible light utilization is generally accompanied by a decreased driving force of the photogenerated carriers to make charge separation extremely difficult. Furthermore, the construction of OWS systems faces serious challenges originating from the sluggish kinetics of water oxidation involving uphill energy barrier and multiple electron transfer[13]. Consequently, visible-light-driven photocatalytic OWS systems are not only limited in number, but also show lower efficiency than those driven by ultraviolet light[14–19]. Accordingly, it is highly desirable to precisely design and modify photocatalysts with efficient visible light utilization for promotion of water oxidation.

N-type monoclinic bismuth vanadate ($BiVO_4$) has emerged as one of the most promising visible-light-responsive photocatalysts and photoanodes for water oxidation since Kudo's report in 1998[20–28]. Owing to its advantages, such as efficient light absorption in the visible light region, good carrier mobility, controllable exposed facets, and non-toxic properties, $BiVO_4$ semiconductor has been widely and successfully employed as the water oxidation photocatalyst for the assembly of Z-scheme OWS systems using solid conductor (i.e., Au, reduced graphene oxide) or redox couple (i.e., $Fe^{3+/2+}$, $[Fe(CN)_6]^{3-/4-}$) as electron mediator[15,16,29–31]. Specifically, our previous work revealed that the spatial separation of photogenerated electrons and holes can be achieved on the anisotropic facets of $BiVO_4$[32], based on which reduction and oxidation cocatalysts are selectively deposited on different facets to remarkably promote its water oxidation and the efficiency of OWS under visible light[16]. Although the $BiVO_4$ photocatalyst has been widely investigated for the assembly of artificial Z-scheme OWS systems, the apparent quantum efficiency (AQE) and STH conversion efficiency achieved so far are still considerably below what is expected. This is mainly due to the shortage of effective cocatalyst regulation and the lack of in-depth understanding of the microscopic mechanisms behind it[33,34]. Notably, for the assembly of the redox couple-mediated Z-scheme OWS system shown in Fig. 1, the loading of effective cocatalysts is extremely important not only for the acceleration of interfacial electron transfer between the

$H_2$-evolving photocatalyst (HEP) and the $O_2$-evolving photocatalyst (OEP), but also for the promotion of surface reaction kinetics of water splitting[35–38]. Therefore, it is a long-term task to develop innovative cocatalysts and unravel their structures as well as influence mechanism on water splitting.

In this study, we address the sluggish water oxidation of $BiVO_4$ via in situ photodeposition of dual innovative cocatalysts, with emphasis on elucidating the local structures of the cocatalysts and the mechanism of promotion of water oxidation. We demonstrate that the nanocomposite of FeOOH and CoOOH (denoted as $FeCoO_x$) in situ formed on the {110} facet of $BiVO_4$ not only lowers Gibbs free energy barrier of water oxidation, but also makes a better promotion on the electron transfer as well as charge separation compared with the commonly used $CoO_x$ cocatalyst. Furthermore, the Ir cocatalyst in situ deposited on the {010} facet of $BiVO_4$ was found to exhibit superior reduction ability of $[Fe(CN)_6]^{3-}$ ions to our previously reported Au. Based on the facet-selective loading of the innovative dual-cocatalysts, the evolution rate of $O_2$ on the $BiVO_4$ was significantly enhanced, and a particulate Z-scheme OWS system with an AQE of 12.3% at 420 nm and a STH of 0.6%, was finally fabricated using $[Fe(CN)_6]^{3-/4-}$ as a redox mediator and $ZrO_2/TaON$ or $MgTa_2O_{6-x}N_y/TaON$ as the HEP. Our results demonstrate the importance and effectiveness of developing suitable cocatalysts for enhancing interfacial charge separation and surface water oxidation kinetics in promoting solar energy conversion.

## Results

**Structural characterizations of cocatalysts photodeposited.** The anisotropic $BiVO_4$ with exposed {010} and {110} facets was prepared according to our previous study[39]. The Ir nanoparticles and $FeCoO_x$ nanocomposite were in situ photodeposited on the surface of $BiVO_4$ from an aqueous solution containing the precursors $K_2IrCl_6$, $CoSO_4$, and redox $[Fe(CN)_6]^{3-}$ ions. The as-obtained sample is hereafter denoted as Ir-$FeCoO_x$/$BiVO_4$. As expected from our previous findings on the spatial separation of photogenerated electrons and holes on the anisotropic $BiVO_4$[32], the Ir nanoparticles, and $FeCoO_x$ nanocomposite were clearly observed to be selectively deposited on the {010} and {110} facets of $BiVO_4$, respectively (Fig. 2a, b). For comparison, the in situ photodeposition of single Ir or $CoO_x$ particles on $BiVO_4$ was similarly obtained (denoted as Ir/$BiVO_4$ and $CoO_x$/$BiVO_4$), and the sample was characterized by field-emission scanning electron microscopy (FESEM) to further confirm the facet-selective deposition (Supplementary Fig. 1). It should be noted that the

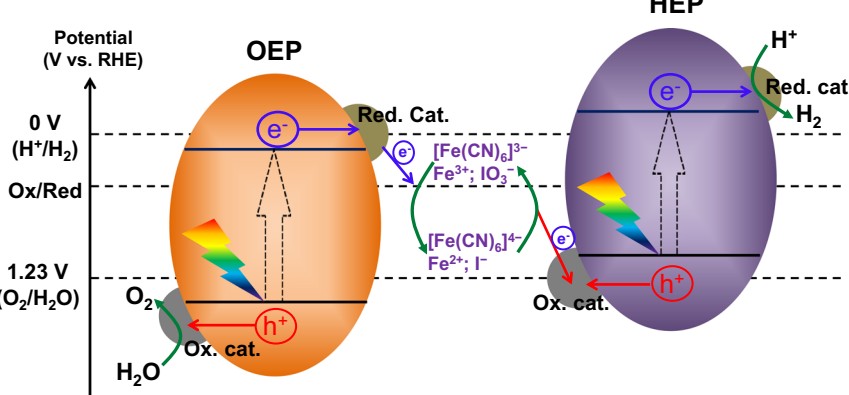

**Fig. 1 The energy diagram for a two-step photoexcitation (also called Z-scheme) system with an aqueous redox mediator for overall water splitting.** Red. cat.: reduction cocatalyst; $O_x$. cat.: oxidation cocatalyst; RHE: reversible hydrogen electrode; HEP: $H_2$-evolving photocatalyst; OEP: $O_2$-evolving photocatalyst.

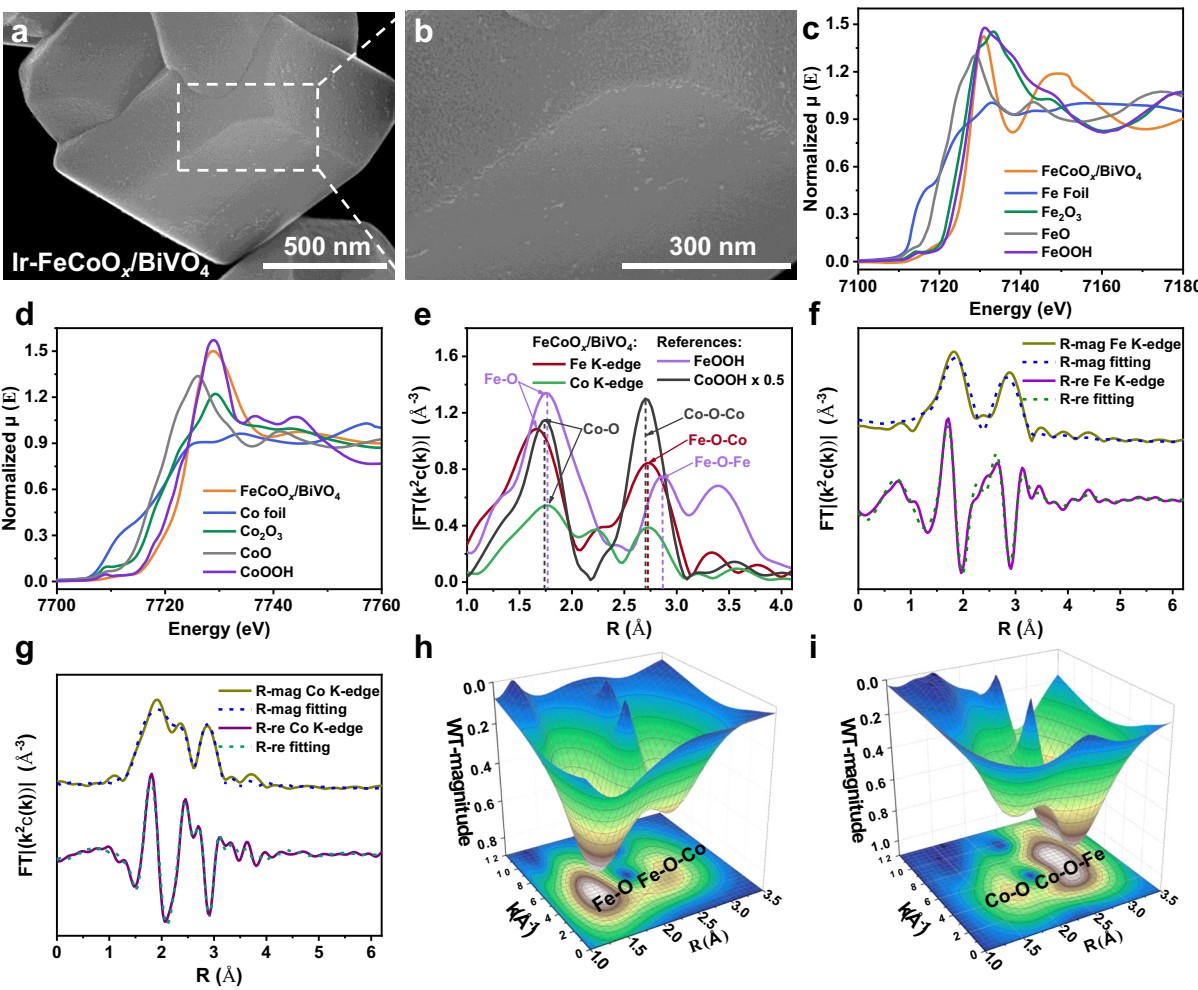

**Fig. 2 Morphology and structural characterizations of typical samples. a**, **b** FESEM images of the Ir-FeCoO$_x$/BiVO$_4$ with different magnification times. **c–i** Structural characterizations of the FeCoO$_x$/BiVO$_4$ sample together with references: **c** Normalized Fe K-edge XANES μ(E) spectra. **d** Normalized Co K-edge XANES μ(E) spectra. **e** Fe K-edge and Co K-edge radial distance χ(R) space spectra. **f** Fourier-transformed (FT)-Extended X-ray absorption fine structure (EXAFS) fitting curves at R space of Fe K-edge. **g** FT-EXAFS fitting curves at R space of Co K-edge. **h** Fe K-edge 3D contour wavelet transform. **i** Co K-edge 3D contour wavelet transform.

morphology of the cocatalysts located on the {110} facet of Ir-FeCoO$_x$/BiVO$_4$ (Fig. 2b) is clearly different from that of the CoO$_x$/BiVO$_4$ sample (Supplementary Fig. 1b), demonstrating the possible interaction between Fe and Co-based compounds. And the change in the long wavelength range of UV-Vis diffuse reflectance spectra (DRS) can confirm the successful deposition of the dual-cocatalysts (Supplementary Fig. 2). The deposited Ir species were verified to exist as metallic Ir nanoparticles by means of X-ray absorption near edge structure (XANES) spectroscopy (Supplementary Fig. 3) and high-resolution transmission electron microscopy image (Supplementary Fig. 4).

To unravel the formation of the FeCoO$_x$ nanocomposite on the {110} facets of BiVO$_4$, the existing state and dispersion of both Fe and Co elements on FeCoO$_x$/BiVO$_4$ (free of Ir nanoparticles to rule out its possible interference during characterization) were first analyzed. According to the elemental mapping results shown in Supplementary Fig. 5, both Fe and Co species are similarly located and dispersed, accompanied by the existence of O, which further demonstrates that Co and Fe combine together in the form of oxidation state during the photo-oxidation process. The deposition of Fe should result from redox ions in the reaction solution. The coexistence of both Fe and Co can be further revealed by electron energy loss spectroscopy analysis

(Supplementary Fig. 6). And their oxidation states can be confirmed to be Fe$^{3+}$ and Co$^{3+}$ by the Fe and Co K-edge XANES measurements through comparing with the reference materials (Fig. 2c, d, respectively).

Second, the radial distance space spectra χ(R) of Fe and Co in FeCoO$_x$/BiVO$_4$ and their corresponding references were analyzed, which provides more convincing support for the formation of nanocomposite. As shown in Fig. 2e and Supplementary Fig. 7, the peaks located at approximately 2.72 Å assigned to the Fe–O–Co bond are consistently observed in both the Fe and Co K-edge of the FeCoO$_x$/BiVO$_4$ sample, but no scattering path signals attributing to the Co–Co bond (2.41 Å) from Co foil, Fe–Fe bond (2.47 Å) from Fe foil, Co–O–Co bond (2.69 Å) from CoOOH, or Fe–O–Fe bond (2.86 Å) from FeOOH can be observed. This clearly reveals that the formation of nanocomposite is a homogeneous phase of bimetallic hydroxide, instead of single-phase Fe or Co hydroxides. It should be pointed out that the possible nanocomposite of Fe$_2$O$_3$ and Co$_2$O$_3$ can be ruled out by comparing the fingerprint feature pattern of normalized XANES μ(E) spectra (Fig. 2c, d and Supplementary Fig. 8a, b) and the first derivative of the normalized XANES μ(E) spectra (Supplementary Fig. 8c, d). In particular, as shown in Supplementary Fig. 8c, d, the peak positions of FeCoO$_x$/BiVO$_4$ are closer

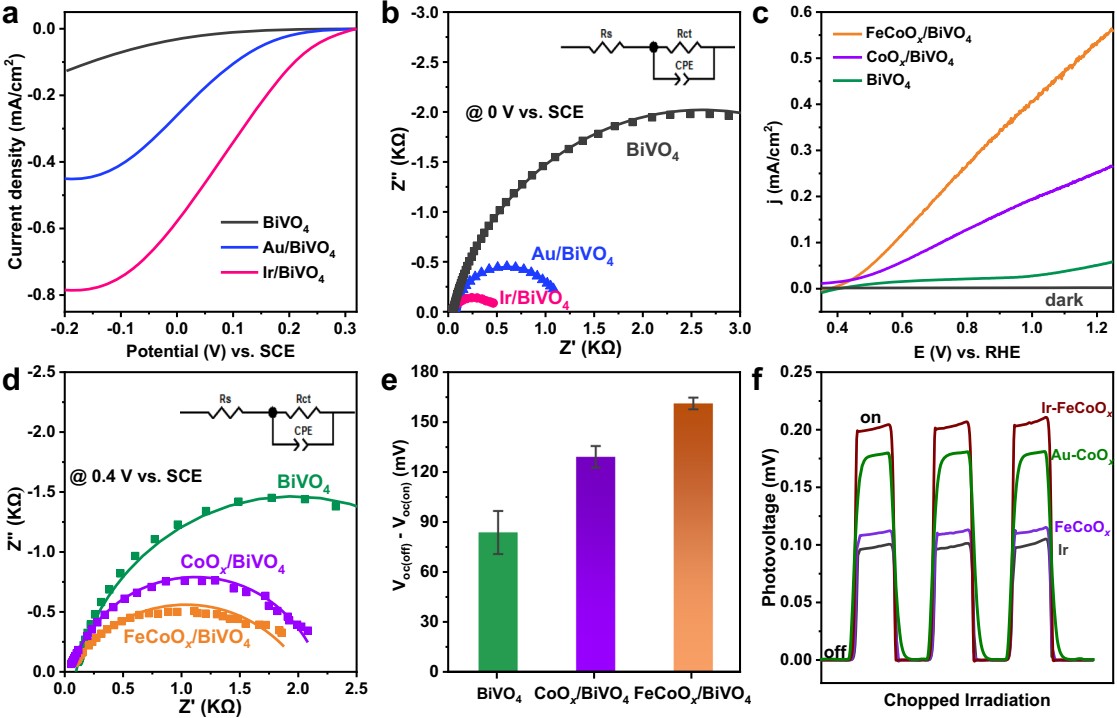

**Fig. 3 Electrochemical measurements and characterizations of typical samples. a** Linear sweep voltammetry curves of typical samples in the 100 mM sodium phosphate buffer solution (pH 6.0) containing 5 mM $K_3[Fe(CN)_6]$. **b** EIS spectra of typical samples in a 100 mM sodium phosphate buffer solution (pH 6.0) containing 5 mM $K_3[Fe(CN)_6]$. SCE, saturated calomel electrode. **c** Photocurrent density-potential curves of $BiVO_4$, $CoO_x/BiVO_4$, and $FeCoO_x/BiVO_4$. **d** EIS spectra of $BiVO_4$, $CoO_x/BiVO_4$, and $FeCoO_x/BiVO_4$ in a 100 mM sodium phosphate buffer solution (pH 6.0). **e** Comparison of difference of OCPs on the $BiVO_4$, $CoO_x/BiVO_4$, and $FeCoO_x/BiVO_4$ under dark and illumination conditions. Measurements were taken at least three times for separate samples and average values are presented with the standard deviation as the error bar. **f** Comparison of promotion effect of cocatalysts on the SPV values of the $BiVO_4$ photocatalyst under chopped visible light irradiation.

to the FeOOH and CoOOH references. Based on these results, the phase species of the $FeCoO_x$ on the surface of $BiVO_4$ sample can be deduced to be more similar to FeOOH/CoOOH with respect to $Fe_2O_3/Co_2O_3$. In addition, compared with the corresponding single-phase hydroxides FeOOH and CoOOH, $Fe/BiVO_4$ sample exhibits much shorter Fe–O bond and longer Co–O bond, and the length of Co–O–Fe bond is between Co–O–Co and Fe–O–Fe (Fig. 2e). This demonstrates the existence of electron transfer and a strong interaction between Fe and Co in the $FeCoO_x/BiVO_4$ sample, providing further proof about the formation of the nanocomposite.

Third, the formation of the nanocomposite can be further verified by the results of quantitative $\chi(R)$ space spectra fitting and wavelet transform of $\chi(k)$. As seen in Supplementary Table 1, Fe–O–Co bond with similar coordination numbers (Fe–O–Co: 2 at ca. 2.745 Å in Fe K-edge; Co–O–Fe: 2 at ca. 2.761 Å in Co K-edge) can be confirmed. The good fitting results of $\chi(R)$ and $\chi(k)$ space spectra (Fig. 2f, g and Supplementary Fig. 9) with reasonable R-factors and the obtained fitting parameters (Supplementary Table 1) provide a quantitative illustration of the existence of Fe–O–Co bond originating from the nanocomposite structure. As similarly revealed in Fig. 2h, i, the Fe–O–Co bond located at [$\chi(k)$, $\chi(R)$] of [4.2, 2.74] or Co–O–Fe bond ([6.4, 2.76]) as well as the Fe–O bond ([4.8, 1.64]) or Co–O bond ([4.2, 1.88]) with two scattering path signal can be observed for both Fe and Co K-edge wavelet transform of $\chi(k)$ spectra of $FeCoO_x/BiVO_4$, but the characteristic scattering path signal of Fe–Fe bond ([8.4, 2.52]), Co–Co bond ([7.8, 2.42]), Fe–O–Fe bond ([5.6, 2.82]) or Co–O–Co bond ([6.8, 2.78]) is not observed as similarly as the reference sample (Supplementary Fig. 10).

**Effect of reduction and oxidation cocatalysts**. As shown in Fig. 1, the water oxidation process of OEP is strongly dependent on both the reduction and oxidation cocatalysts. Therefore, understanding the effect of deposited Ir and $FeCoO_x$ cocatalysts is highly desirable. As depicted in Fig. 3a, the ability of the deposited metallic Ir to reduce $[Fe(CN)_6]^{3-}$ ions was evaluated and found to exhibit a much higher cathode current than that of our previously reported Au nanoparticles on $BiVO_4$[16], indicating its superior performance in activating and reducing the $[Fe(CN)_6]^{3-}$ ions. In addition, the deposition of Ir or Au cocatalyst on the surface of $BiVO_4$ can significantly decrease the charge-transfer resistance ($R_{ct}$) across the semiconductor/electrolyte interface (Fig. 3b), further revealing the effectiveness of the deposited cocatalysts in accelerating the electron transfer from $BiVO_4$ to the $[Fe(CN)_6]^{3-}$ ions (values of $R_s$ and $R_{ct}$ listed in Supplementary Table 2). Meanwhile, the promotion effect of Ir is better than that of Au.

To determine the effect of the $FeCoO_x$ nanocomposite, the efficiencies of charge separation and injection (denoted as $\eta_{sep}$ and $\eta_{inj}$, respectively) on the $FeCoO_x/BiVO_4$ photoanode ($CoO_x/BiVO_4$ and $BiVO_4$ as references) were evaluated by referring to a previous photoelectrochemical analysis[22]. Figure 3c and Supplementary Fig. 11a show that the current of the $BiVO_4$ photoanode can be remarkably promoted after the deposition of $FeCoO_x$ and $CoO_x$ in both cases, with and without the use of a hole scavenger, among which $FeCoO_x$ exhibits a much better promotion effect than $CoO_x$. On this basis, both $\eta_{sep}$ and $\eta_{inj}$ on the $FeCoO_x/BiVO_4$ photoanode were calculated to be higher than that of the $CoO_x/BiVO_4$ photoanode (Supplementary Fig. 11c, d), demonstrating the better promotion effect of the $FeCoO_x$

nanocomposite on both the separation of photogenerated carriers and the injection of holes into the reaction solution (i.e., surface reaction) with respect to $CoO_x$. The excellent promotion of $FeCoO_x$ on the surface reaction can be further supported by the electrochemical impedance spectroscopy (EIS) results given in Fig. 3d and Supplementary Table 3, based on which the $R_{ct}$ resistance on the $FeCoO_x/BiVO_4$ electrode is the smallest among the three electrodes investigated. On the other hand, the superior promotion effect of $FeCoO_x$ on the charge separation can also be evidenced by its larger open-circuit potential (OCP) on the $FeCoO_x/BiVO_4$ compared with $CoO_x/BiVO_4$ (Fig. 3e). Based on the previous result that a larger difference of OCPs under dark and illumination conditions corresponds to more intense band bending[40], therefore the $FeCoO_x/BiVO_4$ sample can be deduced to own a more intense band bending than the $CoO_x/BiVO_4$ sample, leading to a significantly improved $\eta_{sep}$ and the more intense band bending should result from the p-n heterojunction between $FeCoO_x$ and $BiVO_4$[41].

Encouraged by the understanding of the functionalities of both reduction and oxidation cocatalysts (i.e., Ir and $FeCoO_x$), the synergistic effect of dual-cocatalysts on the charge separation was examined using the surface photovoltage (SPV) spectrum. As shown in Fig. 3f, the sample with both Ir and $FeCoO_x$ deposited exhibits a greater SPV amplitude with respect to the sample with single Ir or $FeCoO_x$ loaded. It should be mentioned that a much better promotion effect is also observed for the sample with facet-selective deposition of Ir and $FeCoO_x$ compared to that with facet-selective deposition of Au and $CoO_x$. These results reveal the importance of both facet-selective deposition of dual-cocatalysts and the development of innovative cocatalysts for maximizing the promotion effect.

**Density functional theory calculations on the $O_2$-evolving reaction**. Density functional theory (DFT) calculations were performed to further elucidate the microscopic mechanism of the promotion effect of the $FeCoO_x$ cocatalyst on the $O_2$-evolving reaction (OER) from the viewpoint of both surface catalysis and interfacial charge transfer. As shown in Fig. 4a–c, the $CoO_x$–$FeO_x$–$CoO_x$–$FeO_x$ and $CoO_x$–$CoO_x$–$CoO_x$–$CoO_x$ clusters were simply extracted and placed on the {110} facets of $BiVO_4$ to simulate the $FeCoO_x/BiVO_4$ and $CoO_x/BiVO_4$ interfaces, respectively, which origin from the structure of EXAFS measurement. And the schematic of the whole OER mechanism on the $FeCoO_x/BiVO_4$ and $CoO_x/BiVO_4$ is given in Supplementary Fig. 12 and illustrated in detail in supporting information. Fig. 4d and e presents the Gibbs free energy change diagram of the four elementary steps of OER on the surface of $FeCoO_x/BiVO_4$ and $CoO_x/BiVO_4$, during which the Co and Fe sites on $FeCoO_x/BiVO_4$ were mainly considered as the active sites, respectively. It was demonstrated that the rate-determining step of $FeCoO_x/BiVO_4$ (Co or Fe site) and $CoO_x/BiVO_4$ is the adsorption of one $OH^-$ to form OOH* from O*. The largest decrease of the Gibbs free energy barrier was observed for $FeCoO_x/BiVO_4$ (Co site), whereas the OER performance of $FeCoO_x/BiVO_4$ (Fe site) is much weaker than that of the corresponding Co site, suggesting that the Co site acts as the main OER site.

Next, we plotted the calculated densities of states (DOS) of the $BiVO_4$ {110} surface, $CoO_x/BiVO_4$ {110} interface, and $FeCoO_x/BiVO_4$ {110} interface (Fig. 4f, g, h and Supplementary Fig. 13). Both $CoO_x$ and $FeCoO_x$ are set to be located at the {110} facets of $BiVO_4$. For the bare $BiVO_4$ {110} surface, there is a direct bandgap about 2.1 eV between the valence band and conduction band (Fig. 4f). However, when the $CoO_x$–$FeO_x$–$CoO_x$–$FeO_x$ cluster is settled on the {110} surface of $BiVO_4$, a mixed band mainly composed of Co 3d, Fe 3d, and O 2p states emerges

between the valence band and conduction band (Fig. 4g). It has been demonstrated that the localization of photoexcited holes, as well as subsequent charge separation can be promoted through the formation of mixed bands[42]. In addition, the DOS of the $CoO_x/BiVO_4$ {110} interface (Fig. 4h) has no similar result as that of the $FeCoO_x/BiVO_4$ {110} interface (Fig. 4g, bandgap = 2.0 eV), implying that the loading of $FeCoO_x$ on $BiVO_4$ should have better charge separation. In order to microscopically understand the better electron transfer on the $FeCoO_x$ with respect to the $CoO_x$, their bader charges were calculated and compared. As given in Supplementary Table 4, the changing trend of bader charge on the Co active site after introduction of Fe (increase from 1.2 a.u. in $CoO_x/BiVO_4$ to 1.3 a.u. in $FeCoO_x/BiVO_4$) is in line with the changing one of experimental valence state (Supplementary Fig. 14). Compared to the $CoO_x/BiVO_4$, the higher bader charge on the Co active site in $FeCoO_x/BiVO_4$ indicates its stronger oxidation capacity as well as more beneficial electron transfer[43]. In addition, as shown in Supplementary Fig. 15, the d-band center (Ed) value of Co active sites in $FeCoO_x/BiVO_4$ was calculated as $-1.63$ eV, which is sharply increased with respect to the $CoO_x/BiVO_4$ ($-2.56$ eV). This demonstrates that the electronic structure of Co active sites can be well modulated and optimized in the $FeCoO_x/BiVO_4$ due to the introduction of Fe atoms to get much stronger adsorption properties to the OER intermediates according to the d-band center theory[44,45]. According to previous experimental and theoretical demonstration, the Fe site is relatively inactive during the OER process[46,47]. So we deduce that the role of Fe is to assist in modifying the geometric and electronic structure of Co in the OER together with our results that very limited contributions of Fe 3d states are observed for the mixed band (Fig. 4g). These conclusions from DFT calculation well match with the aforementioned experimental results.

**Photocatalytic performances of Z-scheme OWS**. The modified $BiVO_4$ was employed as an OEP for the assembly of efficient Z-scheme OWS systems together with $ZrO_2/TaON$ or $MgTa_2O_{6-x}N_y/TaON$ as a HEP under visible light irradiation. The HEPs were prepared and modified with cocatalysts according to previously reported procedures[16,48,49], and the diffraction structure and morphology features were coarsely revealed by their powder X-ray diffraction patterns and FESEM images (Supplementary Figs. 16, 17). The contents of deposited Ir and Co on $BiVO_4$ were optimized to be 0.8 and 0.2 wt%, respectively, via the photocatalytic $O_2$ evolution reaction (Supplementary Figs. 18, 19). As seen in Fig. 5a, stable evolution curves of $H_2$ and $O_2$ with the stoichiometric molar ratio of 2:1 can be observed at the experimental region using the optimized photocatalysts, indicating the successful achievement of the OWS process. Moreover, regardless of using $ZrO_2/TaON$ or $MgTa_2O_{6-x}N_y/TaON$ as the HEP, similar OWS activities with the initial rates of $H_2$ and $O_2$ evolution (ca. 160 and 80 μmol/h, respectively) were separately observed, implying that the $O_2$ evolution on $BiVO_4$ is the rate-determining step, as similarly observed in our previous study[16]. It should be pointed out that when the Ir-$CoO_x$(Imp.)/$BiVO_4$ with Ir and Co randomly impregnated is employed as the OEP, the OWS will not be achieved owing to the significantly decreased $O_2$-evolving activity (Supplementary Fig. 20). This indicates the importance of facet-selective deposition of dual-cocatalysts in promoting the $O_2$-evolving activity and fabricating a successful OWS system. The multiple cycles of time-course curves shown in Supplementary Figs. 21 and 22 demonstrate the good photostability of the system constructed in this study. The AQE value of OWS as a function of absorption wavelength was found to be in good accordance with the UV-Vis DRS of the OEP and HEP,

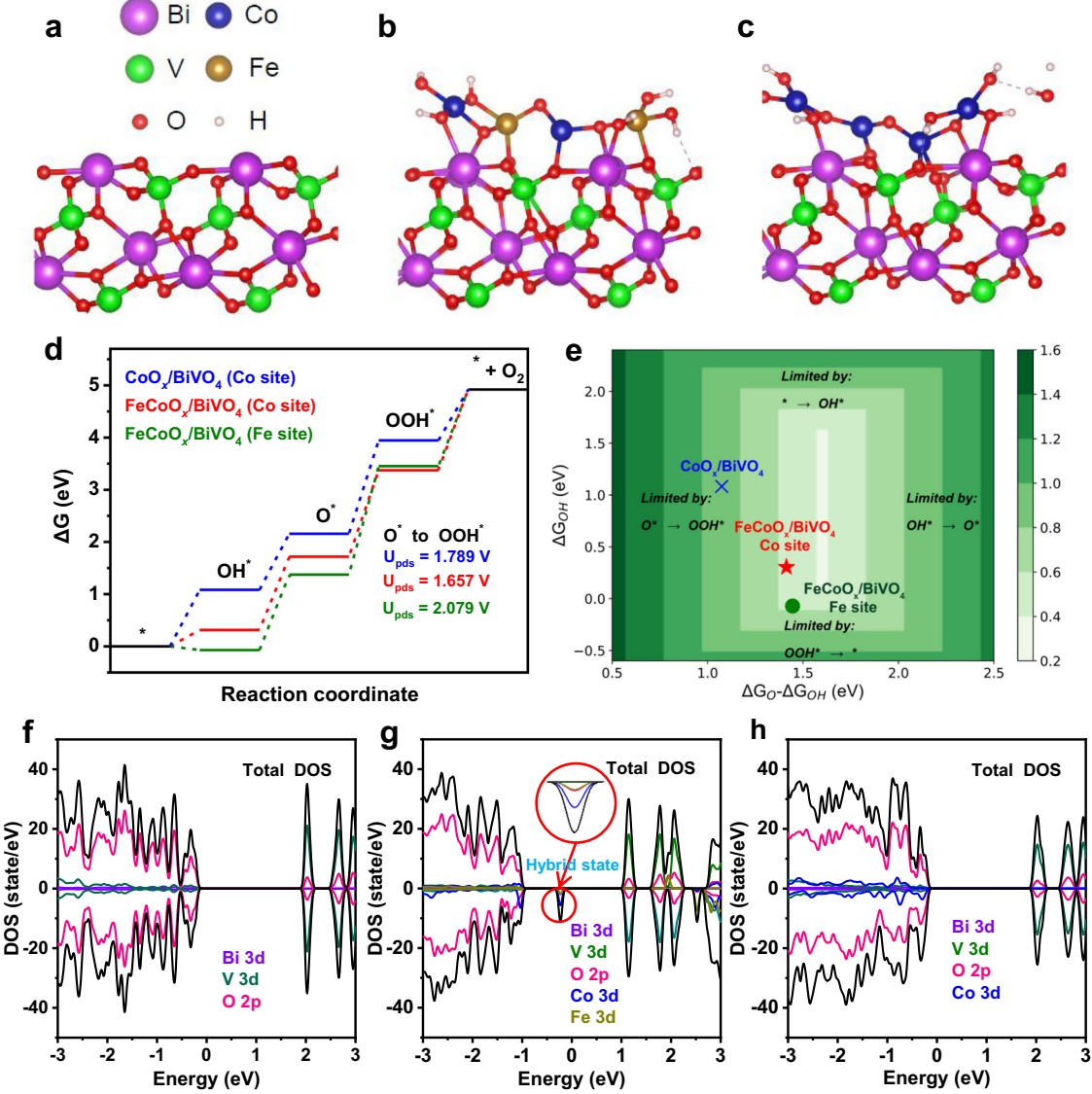

**Fig. 4 Theoretical understanding of the promotion effect of the FeCoO$_x$ cocatalyst.** Visual representation of structures of BiVO$_4$ {110} surface (**a**) FeCoO$_x$/BiVO$_4$ {110} interface (**b**) and CoO$_x$/BiVO$_4$ {110} interface (**c**) for the DFT calculations. **d** Free energy diagram for OER process on FeCoO$_x$/BiVO$_4$ and CoO$_x$/BiVO$_4$ {110} interfaces. The surface structures with various reaction intermediates are shown alongside the free energy diagram. U$_{pds}$, equilibrium potential for the potential determining step. **e** Theoretical overpotential plot with O* and OH* binding energies as descriptors. Calculated densities of state for the BiVO$_4$ {110} surface (**f**), FeCoO$_x$/BiVO$_4$ {110} interface (**g**), and CoO$_x$/BiVO$_4$ {110} interface (**h**).

indicating that the Z-scheme OWS system is driven by visible light excitation (Fig. 5b and Supplementary Fig. 23). The optimal AQE value of OWS at 420 ± 10 nm is 12.3%, and the AQE value at the 500 ± 10 nm is about 3%, demonstrating the wide visible light utilization. According to the activity measurements under the irradiation of AM 1.5 G (Fig. 5c), the STH energy conversion efficiency was calculated to be 0.6%. To the best of our knowledge, both the AQE and STH values should be the highest among the suspending particulate photocatalytic OWS systems using inorganic semiconductor materials with visible light utilization, regardless of one-step or two-step (i.e., Z-scheme) systems.

## Discussion
Here, we show a highly efficient Z-scheme OWS system with benchmarked AQE and STH value over particulate inorganic semiconductor photocatalysts with visible light utilization. The success is mainly ascribed to the in situ facet-selective photo-deposition of innovative dual-cocatalysts (Ir nanoparticles and

FeCoO$_x$ nanocomposite), based on which the sluggish water oxidation on BiVO$_4$ can be largely overcome. Besides the finding and structural unraveling of efficient cocatalysts, the microscopic work mechanism of both reduction and oxidation cocatalysts on the interfacial charge transfer and surface catalysis has been well elucidated respectively. These results should be encouraging and enlightening to the design and assembly of OWS systems for more efficient solar-to-chemical energy conversion.

## Methods
**Synthesis of modified-TaON and BiVO$_4$.** ZrO$_2$-modified TaON (Zr/Ta = 0.1) sample and MgTa$_2$O$_{6-x}$N$_y$/TaON (Mg/Ta = 0.15) composite were used as the HEPs. The ZrO$_2$-modified sample was synthesized by nitridation of the ZrO$_2$/Ta$_2$O$_5$ composite and the MgTa$_2$O$_{6-x}$N$_y$/TaON was prepared by nitridation of the MgTa$_2$O$_6$/Ta$_2$O$_5$ composite under an ammonia flow (20 mL min$^{-1}$) at 1123 K for 15 h by referring to the previous works[48,49]. BiVO$_4$ was chosen as the OEP, which was similarly synthesized according to our previous hydrothermal process[39]. Typically, 10 mmol NH$_4$VO$_3$ and 10 mmol Bi(NO$_3$)$_3$·5H$_2$O were dissolved in 2.0 M nitric acid solution, whose pH value was then adjusted to be about 0.5 with

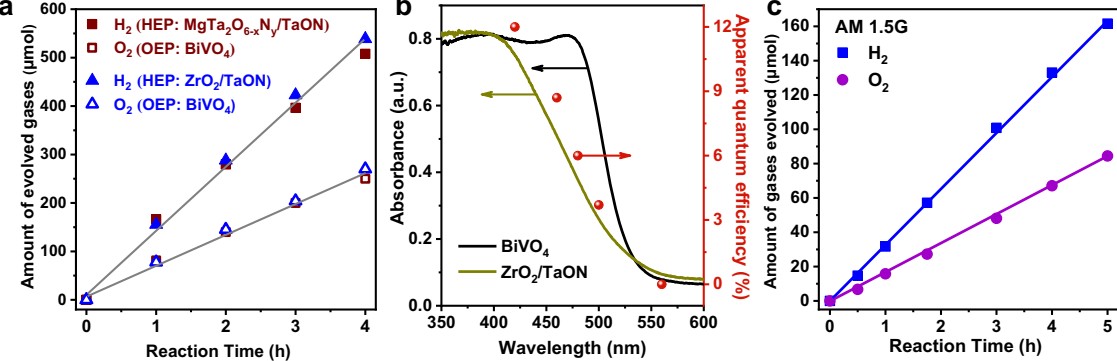

**Fig. 5 Photocatalytic activity of Z-scheme OWS. a** Time course of Z-scheme OWS on the optimized conditions under visible light irradiation. **b** Dependence curve of AQE value as a function of irradiation wavelength, and UV-Vis DRS of the HEP and OEP. **c** Time curve of Z-scheme OWS under illumination of the standard solar simulator (AM 1.5 G, 100 mW cm$^{-2}$). Reaction conditions: **a** 50 mg OEP, 50 mg HEP (ZrO$_2$/TaON, 1.0 wt% Rh, 1.5 wt% Cr) or 50 mg OEP, 100 mg HEP (MgTa$_2$O$_{6-x}$N$_y$/TaON, 2.5 wt% Rh, 3.75 wt% Cr), 100 mL 25 mM sodium phosphate buffer solution (pH 6.0) containing K$_4$[Fe(CN)$_6$] (10 mM), 300 W xenon lamp (λ ≥ 420 nm), temperature: 288 K, Pyrex top-irradiation type. **b** 75 mg OEP, 75 mg HEP (ZrO$_2$/TaON, 1.0 wt% Rh, 1.5 wt% Cr), 150 mL 25 mM sodium phosphate buffer solution (pH 6.0) containing K$_4$[Fe(CN)$_6$] (10 mM), 300 W xenon lamp, temperature: 298 K, Pyrex top-irradiation type. (c) 50 mg OEP, 50 mg HEP (ZrO$_2$/TaON, 1.0 wt% Rh, 1.5 wt% Cr), 100 mL 25 mM sodium phosphate buffer solution (pH 6.0) containing K$_4$[Fe(CN)$_6$] (10 mM), temperature: 288 K, Pyrex top-irradiation type.

ammonia solution (25–28 wt%). The mixed solution was strongly stirred until the observation of a light yellow precipitate that was further aged for about 2 h and then transferred to a Teflon-lined stainless steel autoclave for 10 h hydrothermal treatment at 473 K.

**Preparation of Ir/BiVO$_4$ and CoO$_x$/BiVO$_4$.** The deposition of Ir or CoO$_x$ on the surface of BiVO$_4$ was carried out by the photodeposition method. Typically, 0.2 g BiVO$_4$ powder was dispersed in deionized water containing a calculated amount of K$_2$IrCl$_6$ (2.0 wt%) or CoSO$_4$ (2.0 wt%), and hole (CH$_3$OH) or electron (NaIO$_3$) scavenger, separately. The well-mixed solution was then irradiated by 300 W xenon lamp free of any cut-off filter for 2 h. The as-obtained powders after filtration and washing are correspondingly denoted as Ir/BiVO$_4$ and CoO$_x$/BiVO$_4$, which were used for further characterizations and tests.

**Preparation of FeCoO$_x$/BiVO$_4$ and Ir-FeCoO$_x$/BiVO$_4$.** Both of the samples were similarly prepared by the in situ photodeposition. Meanwhile, 25 mM phosphate buffer solution (PBS, pH = 6, 50 mL) containing a calculated amount of CoSO$_4$ and [Fe(CN)$_6$]$^{3-}$ ions was prepared for the synthesis of FeCoO$_x$/BiVO$_4$, while 25 mM phosphate buffer solution (PBS, pH = 6, 50 mL) containing a calculated amount of K$_2$IrCl$_6$, CoSO$_4$ and [Fe(CN)$_6$]$^{3-}$ ions was prepared for synthesis of Ir-FeCoO$_x$/BiVO$_4$.

**Preparation of HEP.** The deposition of nanoparticulate rhodium-chromium mixed oxides (denoted as Rh$_y$Cr$_{2-y}$O$_3$) as a cocatalyst was carried out by the photo-deposition method. 0.2 g ZrO$_2$-modified TaON or MgTa$_2$O$_{6-x}$N$_y$/TaON was dispersed in 20 v% 150 mL methanol solution. A certain amount of Na$_3$RhCl$_6$ and K$_2$CrO$_4$ (1.0 wt% Rh and 1.5 wt% Cr vs. photocatalyst for ZrO$_2$/TaON and 2.0 wt% Rh and 3.75 wt% Cr vs. photocatalyst for MgTa$_2$O$_{6-x}$N$_y$/TaON) were added as the precursors. The deposition was carried out under the full-spectral irradiation of 300 W xenon lamp for 6 h. Whereafter, the irradiated solution was centrifuged and washed with distilled water, and then dried at 353 K for overnight to get powder for use.

**Preparation of BiVO$_4$ electrodes.** The BiVO$_4$ photoanode was prepared according to the previous work[50]. First of all, Bi(NO$_3$)$_3$·5H$_2$O, NH$_4$VO$_3$, and polyvinyl alcohol were dissolved in 60% HNO$_3$ to prepare the precursor solution. Then the precursor solution was spin-coated on the FTO followed by heat treatment at 623 K for 2 h in air to form the BiVO$_4$ seed layer. Second, the treated FTO was immersed in 2.0 M HNO$_3$ aqueous solution containing Bi(NO$_3$)$_3$·5H$_2$O and NH$_4$VO$_3$, whose pH was adjusted to be 0.9 by adding NH$_3$·H$_2$O drop by drop. The formed BiVO$_4$ precursor film solution was transferred to a Teflon-lined autoclave with the as-prepared substrate for hydrothermal treatment at 473 K for 12 h. The BiVO$_4$ photoanode film was finally calcined at 773 K for 4 h.

As for the selective deposition of Ir and FeCoO$_x$ cocatalysts on the BiVO$_4$ photoanode, similar in situ photodeposition method as the powder was adopted. Specifically, the photoanode was immersed in 25 mM phosphate buffer solution (PBS, pH = 6, 50 mL) containing the K$_2$IrCl$_6$ or/and CoSO$_4$ (K$_2$IrCl$_6$: 40 μL; CoSO$_4$: 10 μL, the concentration of solution: 1 mg/mL) and K$_3$[Fe(CN)$_6$] (0.5 mM) and irradiated for 3 h. Similarly, CoO$_x$ was photodeposited on the surface of BiVO$_4$ to prepare the CoO$_x$/BiVO$_4$ photoanode.

**Measurements of AQE and STH conversion efficiency.** The AQE was measured using a Pyrex top-irradiation-type reaction vessel and a 300 W xenon lamp fitted with band-pass filters (ZBPA420, Asahi Spectra Co., FWHM: 10 nm). The number of photons reaching the solution was measured using a calibrated Si photodiode (LS-100, EKO Instruments Co., LTD.), and the AQE (ϕ) was calculated using the following Eq. (1):

$$\phi(\%) = (AR/I) \times 100 \qquad (1)$$

where $A$, $R$, and $I$ are coefficients, $A$ represents a coefficient (4 for H$_2$ evolution; 8 for O$_2$ evolution) and $R$ represents the evolution rate of H$_2$ or O$_2$. As measured and calculated, the total number of incident photons at the wavelength of 420, 460, 480, 500, and 560 nm are 8.4 × 10$^{20}$, 6.5 × 10$^{20}$, 7.1 × 10$^{20}$, 4.8 × 10$^{20}$, and 6.9 × 10$^{20}$ photons h$^{-1}$, respectively. The evolution rates of H$_2$ on the system containing Rh$_y$Cr$_{2-y}$O$_3$-ZrO$_2$/TaON and Ir-FeCoO$_x$/BiVO$_4$ photocatalysts under the wavelength of 420, 460, 480, 500, and 560 nm were tested to be 41.6, 23.0, 17.5, 7.4, and 0 μmol h$^{-1}$, respectively. The evolution rates of H$_2$ on the system containing Rh$_y$Cr$_{2-y}$O$_3$-MgTa$_2$O$_{6-x}$N$_y$/TaON and Ir-FeCoO$_x$/BiVO$_4$ photocatalysts under the wavelength of 420, 460, 480, and 560 nm were tested to be 42.4, 19.0, 9.0, and 0 μmol h$^{-1}$, respectively.

The STH energy conversion efficiency (η) was calculated according to the following Eq. (2):

$$\eta(\%) = (R_H \times \Delta G^O)/(P \times S) \times 100 \qquad (2)$$

where $R_H$, $\Delta G^\circ$, $P$, and $S$ denote the rate of H$_2$ evolution (mol s$^{-1}$) in photocatalytic water splitting, standard Gibbs energy of water (237.13 × 10$^3$ J mol$^{-1}$), intensity of simulated sunlight (0.1 W cm$^{-2}$), and irradiation area (4.0 cm$^2$), respectively. The light source was an AM 1.5 G solar simulator (XES-40S2-CE, San-Ei Electric), and a top-irradiation reaction vessel was used. The initial rates of H$_2$ and O$_2$ evolution are about 36 and 18 μmol/h, separately.

**Photoelectrochemical tests.** As for the tests of linear sweep voltammetry (LSV) and EIS, a platinum plate was used as a counter electrode and the saturated calomel electrode (SCE) as the reference electrode. The phosphate buffer solution (pH = 6, 0.1 M) with 5 mM K$_3$[Fe(CN)$_6$] aqueous solution and phosphate buffer solution (pH = 6, 0.1 M) were used as the electrolyte. The potential of the working electrode was controlled by a potentiostat (CHI 660E) for the LSV test and potentiostat (Solartron analytic AMETEK) for the EIS test. Before the measurement, the solution was purged with argon gas. The Nyquist plots calculated from EIS were performed from 100,000 to 0.1 Hz. Data were fitted using Zview software.

Current–voltage (J–V) curves under irradiation and darkness were recorded on an electrochemical workstation (CHI 660E). The OCP of photoanode were recorded under illumination and darkness using electrochemical workstation (Solartron analytic AMETEK). A 300 W xenon lamp was used as the light source and the irradiation intensity was high enough to produce a flat band condition of the photoanodes. The electrolyte for J–V curves and OCP was 1 M KBi (pH = 9). 0.2 M Na$_2$SO$_3$ was added to the electrolyte as a hole scavenger if necessary.

## Data availability

The data that support the findings of this study are available from the source data. Source data are provided with this paper.

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

## Acknowledgements

This work was financially supported by the National Natural Science Foundation of China (21902156, 21925206, 21633009), the National Key R&D Program of China (2020YFA0406102, 2017YFA0204904), the DICP Foundation of Innovative Research (DICP I201927), and the Dalian Science and Technology Innovation Fund (2020JJ26GX032). We gratefully acknowledge the BL14W1 beamline of the Shanghai Synchrotron Radiation Facility (SSRF), Shanghai, China, and the 1W1B beamline of the Beijing Synchrotron Radiation Facility (BSRF), Beijing, China for providing the beam time. The numerical calculations in this paper have been done on the supercomputing system in the Supercomputing Center of University of Science and Technology of China.

## Author contributions

F.Z. conceived and designed the experiments. Y.Q. carried out most of the preparations, activity test, catalyst characterizations, and wrote the first draft. J.Z. carried out the XAFS measurements and analysis. Y.K. conducted DFT calculation. Y.Z. assisted the synthesis of BiVO₄ photocatalyst. S.C. assisted the synthesis of MgTa₂O₆-xNᵧ/TaON heterostructure photocatalyst. D.L. assisted the synthesis of BiVO₄ photoanode. W.L. conducted the HRTEM and EELS measurements. Y.C. and T.X. conducted the SPV characterizations. F.Z. and C.L. directed the work and revised the manuscript. J.C. gave corrections about DFT calculation part. K.D. instructed synthesis of H₂-evolving photocatalysts and provided useful suggestions and discussion on the EXAFS results. All authors discussed the results and contributed to the manuscript.

## Competing interests

The authors declare no competing interests.
