## [Peer Review File · Nature Communications]

Unraveling of cocatalysts photodeposited selectively on facets of BiVO₄ to boost solar water splittingREVIEWER COMMENTS

Reviewer #1 (Remarks to the Author):

Report on “Unravelling of innovative dual...”

By Yu Qi et al

I have gone through this m/s very carefully and have the following comments. Significance of Work

The work addresses an important problem of the bottleneck with the 4e oxidation of water to dioxygen (OER) in the overall water photosplitting pathway. I am, however, not a huge fan of particulate “photocatalysts” (the reason for the quotation marks will become clear below) and this was not a flow system such that products would accumulate and cause back reactions---a perennial problem with the particulate approach. Nonetheless, I am willing to look instead at the SCIENCE and not the TECHNOLOGY aspects of this work. In this regard, the innovation was sufficiently high to merit consideration in your esteemed journal BUT for the following reasons.

- 1) The facet-selective deposition of “cocatalysts” (again note the quotes, see below) has been published previously by the same group (see two of the publications cited herein, Refs. 16 and 39, there may be more).
- 2) The mechanistic aspects of the (photo)deposition of the “cocatalysts is not well addressed. WHY do they selectively deposit on the two different facets? If they have described reasons why in these previous publications, then they cannot claim “innovative” here including in the title! EXAMPLE: Lines 149, 150: “The deposition of Fe should result from redox...” Why and how?!
- 3) The MS is TERRIBLY written with poor grammar and syntax. EXAMPLES: No such word in the English language as “composited” (Line 148) and immediately after: the statement “...Co and Fe are composited in an oxidation state...” makes absolutely no sense. Surely, they mean that Fe and Co are not deposited in an elemental state and they are bonded together? Line 64: “...visible light are NOT limited...” (What does NOT mean?) Lines: 86,87: What exactly does cocatalyst modulation mean?”
- 4) The XANES data, while convincing, need to be rewritten completely for better clarity. Right now, it is written for an X-ray spectroscopist (specialist) in rather poor English.
- 5) Is the Fe-Co “cocatalyst” amorphous? Did it show an XRD pattern? Does this solid -state compound exist in the literature? If so, what are its catalytic properties? Do the authors have data on this catalyst located on a glassy carbon surface for the OER?
- 6) I deduce that iron’s role is merely to introduce mid gap states in the compound and thereby mediate ET to and from Co sites. The Co sites themselves are the active sites.

- 7) I have a HUGE problem with the authors' nomenclature: "Photocatalysts", "photocatalytic", "cocatalysts" etc. Is the authors' Z-scheme downhill or uphill? Where is the thermodynamic analysis of the overall and component reaction steps? If uphill, how can the semiconductor be a photocatalyst? If the semiconductor surface itself is not catalytic (and it is NOT, see black curve in Fig. 2a), then use of the word "cocatalyst" is silly and wrong! Simply use "catalyst" etc. and replace all the "photocatalysts" with semiconductors etc. and "photocatalytic" with photoelectrochemical!
- 8) The EIC data are plotted wrongly; the authors MUST use symmetric axes for the real and imaginary components of the Nyquist plot so arcs do not appear distorted!
- 9) Please do not use words like "innovative" in the title!
- 10) What does it mean: JC and KD provided "constructive advice"? Does "advice" suffice to get authorship on a paper in a major journal?

RECOMMENDATION: Reject to resubmit with massive revision. I would want to reexamine how the authors have responded to the above points in a completely revised and BETTER WRITTEN m/s.

Reviewer #2 (Remarks to the Author):

The authors report the design of BiVO₄ with distinct facets with the function of spatial separation of electrons and holes. Based on the redox reaction induced at the specific facet, cocatalysts particles can be selectively deposited on the desired facet of BiVO₄ (i.e. FeCoO_x and Ir). Enhanced OER was observed on the selectively deposited dual cocatalyst loaded BiVO₄. Combining these BiVO₄ oxygen evolving photocatalyst (OEP) with hydrogen evolving photocatalyst (HEP) to construct Z-scheme system, overall water splitting to produce stoichiometric amount of H₂ and O₂ was achieved. High solar-to-hydrogen conversion efficiency was recorded. Although the results and characterisation are decent, this work raises concerns in terms of novelty, innovation as well as some fundamental explanation. The reviewer considers this work not at the expected high level for Nature Communications.

1. Selective (photo) deposition of dual cocatalysts on faceted BiVO₄ or other Bi-based ternary oxides has been widely reported. The authors group is one of the pioneering groups reported this strategy back in 2013. In a more recent work in Joule (10.1016/j.joule.2018.07.029), the authors actually adopted this selective deposition of dual cocatalysts on BiVO₄ (CoO_x and Au, respectively) and constructed a Z-scheme system for overall water splitting. In this work, CoO_x and Au cocatalysts were replaced by FeCoO_x and Ir, respectively). Although the activities reported in this work are improved, the

methodology and guiding principle of this work are not new or sufficiently impactful for Nature Commun.

2. FeCoOx is adopted as the cocatalyst for oxygen evolution in this study. FeCoOx loaded on BiVO4 was first reported in Adv. Funct. Mater. 2018, 28, 1802685. Detailed study of the role and impact of FeCoOx on BiVO4 for oxygen evolution was reported and supported by theoretical modelling. That pioneering work on FeCoOx/BiVO4 is, however, not referred in this manuscript and the reviewer believes it is an important document related to this study.

3. When Ir and FeCoOx were selectively deposited at the desired sites of BiVO4, charge transfer boosted by the cocatalyst were maximized. When randomly deposited, the constructive effect was not maximized, but still should be beneficial in general as compared to the bare BiVO4. The reviewer is curious on why the overall water splitting was not achieved when the cocatalysts were randomly deposited through impregnation. The random location of cocatalyst shouldn't quench totally the Z-scheme reaction.

4. When using different HEP (ZrO2/TaON or MgTa2O6-xNy/TaON), similar OWS (H2 and O2 evolution) activities were obtained. The authors claimed that it proved the OER on BiVO4 to be the rate determining. The reviewer agrees that the rate determining step is the OER on BiVO4. However, OWS through Z-scheme depends on many factors including the interparticulate interaction between OEP and HEP in this work. The similar activities of H2 and O2 evolution under different HEP (ZrO2/TaON or MgTa2O6-xNy/TaON) is not clearly understood in this work. If the authors' argument is correct, the same OWS performance will also be observed when the authors fixed the OEP configuration (selective vs random deposition; with single cocatalyst or dual cocatalyst) while switching between the HEP.

5. From OCP discussion, FeCoOx/BiVO4 has larger band bending than CoOx/BiVO4 and therefore better charge transfer. The authors may want to explain further on why such larger band bending was observed. It was suggested in previous study that FeCoOx forms p-n heterojunction with BiVO4 and promoted the holes transfer for OER.

Reviewer #3 (Remarks to the Author):

The authors demonstrate the enhancement on the water oxidation performance of BiVO4 using novel cocatalysts that have been prepared via in-situ photodeposition. The author also performed various and complementary analyses including DFT and XAS to find out the mechanism of water oxidation

promotion in their system. This manuscript provides credible results and reasonable discussion which is related to the results. I would recommend the publication of this manuscript after addressing the following comments.

1) The authors claimed that the OWS system in this study shows good photostability. However, the stability test time in Supplementary Figure 19 is not enough to support this claim. The authors should provide the stability test for at least more than 50 hours under the illumination of both 300 W xenon lamp ($\lambda \geq 420$ nm) and solar simulator (1 sun).

2) The authors showed only the UV-Vis spectra of Ir-FeCoO_x/BiVO₄ OEP. I'm wondering if there is any difference in the light absorption property of bare BiVO₄ and Ir-FeCoO_x/BiVO₄. The authors should provide the UV-Vis spectra of both OEPs.

Responses to the reviewer(s)' comments

First of all, we really appreciate the referees for spending their valuable time on reviewing our submission. We also thank the editors for giving us the opportunity to improve the quality of our work. To well address the reviewers' concerns, we will give a detailed response point by point as below.

Reviewer #1:

The work addresses an important problem of the bottleneck with the 4e oxidation of water to dioxygen (OER) in the overall water photosplitting pathway. I am, however, not a huge fan of particulate “photocatalysts” (the reason for the quotation marks will become clear below) and this was not a flow system such that products would accumulate and cause back reactions---a perennial problem with the particulate approach. Nonetheless, I am willing to look instead at the SCIENCE and not the TECHNOLOGY aspects of this work. In this regard, the innovation was sufficiently high to merit consideration in your esteemed journal BUT for the following reasons.

1. The facet-selective deposition of “cocatalysts” (again note the quotes, see below) has been published previously by the same group (see two of the publications cited herein, Refs. 16 and 39, there may be more).

Response: We agree with the reviewer that the facet-selective deposition of cocatalysts is not original. Actually, the facet-selective deposition of cocatalyst was first reported by our group (Li, C. et al., *Nat. Commun.*, **2013**, 4: 1432) and has been proved by many other groups using different semiconductors (Guo, L. J. et al., *Nanoscale*, **2014**, 6, 9695; Cheng, H. M. et al., *Chem. Commun.*, **2014**, 50, 10416; Gao, E. P. et al., *Appl. Catal. B Environ.*, **2015**, 162, 470; Domen, K. et al., *Chem. Commun.*, **2015**, 51, 4302; *ACS Appl. Mater. Interfaces*, **2019**, 11, 22264). However, it should be pointed out that the novelty of this work was not claimed by the facet-selective deposition of dual-cocatalysts. Actually, the main innovation in this work is that we explored novel cocatalysts (Ir and FeCoO_x) through the *in situ* photodeposition method to boost the

performance of Z-scheme OWS system instead of the discovery of facet-selective deposition of cocatalysts. It has been well known that the cocatalyst plays a very important role in the photocatalytic system (Li, C. et al., *Acc. Chem. Res.*, **2013**, *46*, 1900). The main functions of cocatalysts are to reduce the overpotential and to improve the charge separation. Over the past decades, a large number of publications have been focused on exploring new cocatalyst, loading methodologies of cocatalysts and functional understanding of structure of cocatalysts for promoted photo(electro)catalytic performances (Li, C. et al., *J. Am. Chem. Soc.*, **2008**, *130*, 7176; *J. Catal.*, **2012**, *290*, 151; Qiao, S. Z. et al., *Chem. Soc. Rev.*, **2014**, *43*, 7787; Choi, K. S. et al., *Science*, **2014**, *343*, 990; Du, P. W. et al., *Energy Environ. Sci.*, **2015**, *8*, 2668; Ida, S. et al., *Angew. Chem. Int. Ed.*, **2018**, *57*, 9073; Piao, L. et al., *Joule*, **2018**, *2*, 549; Chen, X. et al., *Chem. Rev.*, **2019**, *119*, 3962; Hyeon, T., *Nat. Mater.*, **2019**, *18*, 620; Lou, X. et al., *Adv. Mater.*, **2019**, *31*, 1804883; Gong, J. et al., *Adv. Mater.*, **2019**, *31*, 1804710; Domen, K. et al. *Nature*, **2020**, *581*, 411; *Nat. Commun.*, **2021**, *12*: 1005; Wang, X. et al., *Nat. Catal.*, **2020**, *3*, 649; Xing, M et al., *Angew. Chem. Int. Ed.*, **2020**, *59*, 13968; Negishi, Y. et al., *Angew. Chem. Int. Ed.*, **2020**, *59*, 7076; Tanaka, T. et al., *Chem. Sci.*, **2021**, *12*, 4940 etc.). To date, exploitation of novel cocatalyst and understanding of their microscopic underlying mechanism have been being of great importance and significance in the areas of both science and technology, which have become one of the most promising ways to improve photocatalytic performances.

In this work, we found that the nanocomposite of CoOOH and FeOOH (denoted as FeCoO_x) produced through a simple photodeposition method can act as an effective oxidation cocatalyst, and its structure was analyzed by XAFS measurement in detail to be correlated with the performance. What's more, its promotion mechanism on the water oxidation has been well unraveled and discussed. After coupling with the H₂-evolving photocatalyst, the whole Z-scheme OWS activity is well improved. This work not only developed novel cocatalysts to promote apparent quantum efficiency (AQE) of redox-driven particular Z-scheme overall water splitting over inorganic semiconductor powders to reach the record value (12.3% at 420 nm), but also enhanced the scientific understanding to the functionalities and underlying mechanism of both

reduction and oxidation cocatalysts. We do believe that both of them should be encouraging and guiding to broad readership working in the field of photocatalysis, especially for water splitting.

2. The mechanistic aspects of the (photo)deposition of the “cocatalysts is not well addressed. WHY do they selectively deposit on the two different facets? If they have described reasons why in these previous publications, then they cannot claim “innovative” here including in the title! EXAMPLE: Lines 149, 150: “The deposition of Fe should result from redox...” Why and how?!

Response: Thank the reviewer very much for his/her helpful suggestion and discussion. The selective deposition of cocatalysts on two different facets was first reported and well discussed by our previous work (Li, C. et al., *Nat. Commun.*, **2013**, *4*: 1432), and the main reason has been ascribed to the spatial separation of photogenerated electrons and holes among the different facets of BiVO₄. Subsequently, the driving force for charge separation was discussed to mainly result from the different conduction/valence band positions between different facets according to the DFT results (Li, C. et al., *Energy Environ. Sci.*, **2014**, *7*, 1369). What’s more, the Kelvin probe force microscopy and spatial-resolved surface photovoltage spectroscopy have been employed to detect their distinct surface band bendings induced by the surface charge region, as should be responsible for the observed spatial charge separation (Li, C. et al., *Angew. Chem. Int. Ed.*, **2015**, *54*, 9111). However, it should be pointed out again that in this work, the innovation is exploring novel efficient reduction and oxidation cocatalyst instead of the methodology of selective photodeposition of dual-cocatalysts, based on which we fabricated a significantly efficient redox-driven particular Z-scheme overall water splitting with a recorded AQE value of 12.3% at 420 nm. Meanwhile, some advanced techniques together with DFT calculation were employed to well reveal the structure and function of the dual-cocatalysts to deepen the scientific understanding. In this case, we have demonstrated that the FeCoO_x nanocomposites *in situ* photodeposited can work as an innovative and efficient cocatalyst of water oxidation, showing superior performance with respect to previous CoO_x. To follow the reviewer’s suggestion and

avoid any misunderstanding, the title has been slightly modified to be “Unraveling of cocatalysts photodeposited selectively on facets of BiVO₄ to boost solar water splitting”.

From the results of XAFS (Fig. 2c, e, h), elemental mappings images (Supplementary Fig. 5) and EELS spectra (Supplementary Fig. 6), the Fe element truly exists in the FeCoO_x cocatalyst. While, based on the reaction condition, the Fe element only exist in the redox, so it is reasonable to deduce that the deposition of Fe results from redox [Fe(CN)₆]⁴⁻ ions which can be produced from the photoreduction of [Fe(CN)₆]³⁻ ions on the surface of BiVO₄ (see Fig. 1). However, it is still difficult for us to detect and understand the formation mechanism of FeCoO_x nanocomposites because of shortage of operando technology for the photocatalytic system, which is interesting to be investigated in the future but beyond the scope of this work. One possible reason may result from their similar deposition kinetics of Co²⁺ and Fe²⁺ ions and similar structure of FeOOH and CoOOH.

3. The MS is TERRIBLY written with poor grammar and syntax. EXAMPLES: No such word in the English language as “composited” (Line 148) and immediately after: the statement “...Co and Fe are composited in an oxidation state...” makes absolutely no sense. Surely, they mean that Fe and Co are not deposited in an elemental state and they are bonded together? Line 64: “...visible light are NOT limited...” (What does NOT mean?) Lines: 86,87: What exactly does cocatalyst modulation mean?”

Response: Thank the reviewer very much for the helpful suggestion. We have checked the grammar and syntax carefully, and revised the inappropriate expression. The revised part in the revision has been highlighted in yellow background.

4. The XANES data, while convincing, need to be rewritten completely for better clarity. Right now, it is written for an X-ray spectroscopist (specialist) in rather poor English.

Response: Thank the reviewer very much for the helpful suggestion. We have rewritten the discussion about the XANES data. And the revised part in the revision has been highlighted with yellow background (see Page 7-9 of revision).

5. Is the Fe-Co “cocatalyst” amorphous? Did it show an XRD pattern? Does this solid–state compound exist in the literature? If so, what are its catalytic properties? Do the authors have data on this catalyst located on a glassy carbon surface for the OER?

Response: Yes, the FeCoO_x cocatalyst is amorphous according to the results of our following characterizations. On one hand, no obvious lattice fringes are detected in the HRTEM image (Fig. R1). On the other hand, no additional diffraction peaks assigned to the FeCoO_x can be obviously observed by comparing the XRD patterns of BiVO₄ and FeCoO_x/BiVO₄ (Fig. R2). To our knowledge, the FeCoO_x as cocatalyst of the photocatalytic system has not been reported previously. According to our EXAFS results, the phase species of the FeCoO_x can be deduced as the nanocomposite of FeOOH and CoOOH (Fig. 2e, h, i). The detailed discussion and illustration on the formation of composite and its structural understanding can be referred to the first part (*Structural characterizations of cocatalysts in situ photodeposited*) of the results in the revision (Page 6-8).

Concerning the catalytic activity of the FeCoO_x nanocomposite on a glassy carbon surface, it is still difficult to be examined in consideration of the fact that the FeCoO_x in this work was *in situ* photodeposited on the surface of BiVO₄ semiconductor, while the glassy carbon or some other conducting supports such as carbon fiber paper, nickel foam etc. is conductor instead of semiconductor. Accordingly, the similar *in situ* photodeposition cannot happen on the glassy carbon support. However, in order to address the reviewer’s concern about the function of FeCoO_x, we deposited FeCoO_x on the BiVO₄ photoanode for photoelectrochemical water oxidation. As seen in Fig. 3c and Supplementary Fig. 11, the promotion effect of the FeCoO_x nanocomposite on both the separation of photogenerated carriers and the injection of holes into the reaction solution (i.e. surface reaction) is better compared to the conventionally used CoO_x cocatalyst that has been widely proved and employed as cocatalyst to accelerate the water oxidation (Zhang, F. et al., *J. Am. Chem. Soc.*, **2012**, *134*, 8348; Domen, K. et al., *Energy Environ. Sci.*, **2013**, *6*, 3595; Feng, P. et al., *Adv. Mater.*, **2014**, *26*, 5043; Li, C. et al., *Angew. Chem. Int. Ed.*, **2015**, *54*, 3047; Nathan, S. L. et al., *Energy Environ. Sci.*,

2015, 8, 2644; 2016, 9, 892; Osamu, I., et al., *J. Am. Chem. Soc.*, 2016, 138, 14152; Sharp, I. D. et al., *J. Am. Chem. Soc.*, 2017, 139, 8960).

Fig. R1 Representative HRTEM image of Ir-FeCoO_x/BiVO₄.

Fig. R2 XRD patterns of the BiVO₄ and FeCoO_x/BiVO₄.

6. I deduce that iron's role is merely to introduce mid gap states in the compound and thereby mediate ET to and from Co sites. The Co sites themselves are the active sites.

Response: Thank the reviewer very much for his/her helpful suggestion. In the original manuscript, we have expressed that the Fe site is relatively inactive during the OER process and the role of Fe is deduced to assist in modifying the geometric and electronic

structure of Co in the OER (Highlighted in yellow background in Page 14). And the largest decrease of the Gibbs free energy barrier was observed for FeCoO_x/BiVO₄ (Co site), whereas the OER performance of FeCoO_x/BiVO₄ (Fe site) is much weaker than that of the corresponding Co site, suggesting that the Co site acts as the main OER site (Highlighted in yellow background in Page 13) in the discussion of the DFT results.

7. I have a HUGE problem with the authors' nomenclature: "Photocatalysts", "photocatalytic", "cocatalysts" etc. Is the authors' Z-scheme downhill or uphill? Where is the thermodynamic analysis of the overall and component reaction steps? If uphill, how can the semiconductor be a photocatalyst? If the semiconductor surface itself is not catalytic (and it is NOT, see black curve in Fig. 2a), then use of the word "cocatalyst" is silly and wrong! Simply use "catalyst" etc. and replace all the "photocatalysts" with semiconductors etc. and "photocatalytic" with photoelectrochemical!

Response: Thank the reviewer for his/her interesting discussion and suggestions. To address the reviewer's questions, we are very pleased to explain some basic concepts in the field of photocatalysis. First of all, the overall water splitting is uphill reaction regardless of one-step or two-step (i.e. Z-scheme) systems in consideration of the fact that the water oxidation half reaction is uphill reaction involving four electron transfer (Domen, K. et al., *Nat. Rev. Mater.*, **2017**, 2, 17050; *Chem. Soc. Rev.*, **2019**, 48, 2109; Li, C. et al., *Adv. Catal.*, **2017**, 60, 1). To make the thermodynamically uphill reaction (i.e. water splitting) happen, it is basically required that the conduction band of photocatalyst is more negative than the potential of water reduction, while the valence band of photocatalyst should be more positive than the potential of water oxidation. In this case, one semiconductor is necessary to be employed as the photocatalyst to excite the electrons from the valence band to the conduction band. The details on the thermodynamic requirement and basic processes in the Z-scheme system have been given in the Fig. 1 in the original and revised manuscript. The Z-scheme can be divided into two half reactions: one is H₂-evolving reaction and the other is O₂-evolving reaction. This concept was first put forward in 1979 (Bard, A. J., *J. Photochem.*, **1979**,

10, 59) and realized (Arakawa, H., et al., *Chem. Commun.*, **2001**, 2416) under visible light irradiation in 2001.

Secondly, it should be pointed out that the surfaces of the semiconductor photocatalyst are normally active for the water reduction and/or water oxidation, but most of them exhibit low reaction kinetics. That is why the semiconductor can be called as the photocatalyst. To address the extremely low reaction kinetics of semiconductor itself, the loading of one catalyst is necessary and effective for promotion of the reaction kinetics in most of cases. Since the catalyst itself (like some noble metals such as Pt, Rh, Pd etc.) cannot be photo-driven under illumination for catalysis, the catalyst is called as cocatalyst. The functions of the cocatalyst loaded on the surface of semiconductor photocatalyst are to collect photogenerated electrons or holes from the semiconductor and to reduce the reaction overpotential/activation energy of water reduction or water oxidation, respectively (Li, C. et al., *Acc. Chem. Res.*, **2013**, *46*, 1900; Domen, K. et al., *Bull. Chem. Soc. Jpn.*, **2016**, *89*, 627; Zou, J. et al., *Adv. Sci.*, **2019**, *6*, 1801505). For example, the surface of Ta₃N₅ is active but inefficient for the water oxidation, so the loading of cocatalyst has been generally made to promote the charge separation and water oxidation reaction kinetics (Domen, K. et al., *Chem. Lett.*, **2002**, *31*, 736; *Angew. Chem. Int. Ed.*, **2013**, *52*, 11252; *Adv. Mater.*, **2013**, *25*, 125; *Nat. Commun.*, **2013**, *4*: 2566; Zou, Z. et al., *Angew. Chem. Int. Ed.*, **2013**, *52*, 11016; *Adv. Funct. Mater.*, **2012**, *22*, 3066; Li, C. et al., *Angew. Chem. Int. Ed.*, **2015**, *54*, 3047)

Finally, it should be mentioned that the photocatalytic water splitting systems have been widely fabricated, which are normally composed of two parts: one is for the light absorption, and the other is for catalysis. The former is normally composed of the semiconductor, and the latter normally contains the catalyst working in dark. The photocatalytic system as a whole has been habitually called as photocatalyst, and the catalyst is called as cocatalyst. We just followed the traditional habits. Concerning the replacement of photocatalysts with semiconductors, it may be not suitable in consideration of the fact that most of photocatalysts are composed of semiconductor and cocatalyst, and most cocatalysts like noble metals Pt, Rh, Pd etc. are not semiconductors. Concerning the replacement of photocatalytic with

photoelectrochemical, it should be also unsuitable because of the fact that the photocatalytic and photoelectrochemical process completely belong to different routes (Andrew, M. et al., *J. Photochem. Photobiol., A*, **1997**, *108*, 1; Maeda, K. et al, *J. Photochem. Photobiol., C*, **2011**, *12*, 237; Zou, Z. et al., *Energy Environ. Sci.*, **2013**, *6*, 347; Li, C. et al., *Adv. Catal.*, **2017**, *60*, 1; Tang, J. et al., *Chem. Soc. Rev.*, **2017**, *46*, 4645; Domen, K. et al., *Chem. Rev.*, **2020**, *120*, 919).

8. The EIC data are plotted wrongly; the authors MUST use symmetric axes for the real and imaginary components of the Nyquist plot so arcs do not appear distorted!

Response: According to the reviewer's suggestion, the Fig. 3b and 3d have been revised in the revision.

9. Please do not use words like "innovative" in the title!

Response: To follow the reviewer's suggestion, the title has been revised as follows "Unraveling of cocatalysts photodeposited selectively on facets of BiVO₄ to boost solar water splitting".

10. What does it mean: JC and KD provided "constructive advice"? Does "advice" suffice to get authorship on a paper in a major journal?

Response: To follow the reviewer's suggestion, the part of author contributions has been modified with specific illustrations. All the revised parts have been highlighted in yellow background in Page 32 of the revision. Thanks.

Reviewer #2:

The authors report the design of BiVO₄ with distinct facets with the function of spatial separation of electrons and holes. Based on the redox reaction induced at the specific facet, cocatalysts particles can be selectively deposited on the desired facet of BiVO₄ (i.e. FeCoO_x and Ir). Enhanced OER was observed on the selectively deposited dual

cocatalyst loaded BiVO₄. Combining these BiVO₄ oxygen evolving photocatalyst (OEP) with hydrogen evolving photocatalyst (HEP) to construct Z-scheme system, overall water splitting to produce stoichiometric amount of H₂ and O₂ was achieved. High solar-to-hydrogen conversion efficiency was recorded. Although the results and characterisation are decent, this work raises concerns in terms of novelty, innovation as well as some fundamental explanation. The reviewer considers this work not at the expected high level for Nature Communications.

1. Selective (photo) deposition of dual cocatalysts on faceted BiVO₄ or other Bi-based ternary oxides has been widely reported. The authors group is one of the pioneering groups reported this strategy back in 2013. In a more recent work in Joule (10.1016/j.joule.2018.07.029), the authors actually adopted this selective deposition of dual cocatalysts on BiVO₄ (CoO_x and Au, respectively) and constructed a Z-scheme system for overall water splitting. In this work, CoO_x and Au cocatalysts were replaced by FeCoO_x and Ir, respectively). Although the activities reported in this work are improved, the methodology and guiding principle of this work are not new or sufficiently impactful for Nature Commun.

Response: Thank the reviewer for spending his/her valuable time on reviewing our work. Concerning about the novelty of this work, we would like to emphasize that our innovation of this work is exploring novel FeCoO_x and Ir cocatalysts through a simple photodeposition as well as its underlying working mechanism instead of the selective (photo) deposition of dual-cocatalysts. It has been well known that the cocatalyst plays a very important role in promoting photocatalytic activities (Li, C. et al., *Acc. Chem. Res.*, **2013**, *46*, 1900; Domen, K. et al., *Bull. Chem. Soc. Jpn*, **2016**, *89*, 627). The main functions of cocatalysts are to reduce the overpotential and to improve the charge separation. Over the past decades, a large number of publications have been focused on exploring new cocatalyst, loading methodologies of cocatalysts and functional understanding of structure of cocatalysts for promoted photo(electro)catalytic performances (Li, C. et al., *J. Am. Chem. Soc.*, **2008**, *130*, 7176; *J. Catal.*, **2012**, *290*, 151; Qiao, S. Z. et al., *Chem. Soc. Rev.*, **2014**, *43*, 7787; Choi, K. S. et al., *Science*,

2014, 343, 990; Du, P. W. et al., *Energy Environ. Sci.*, **2015**, 8, 2668; Ida, S. et al., *Angew. Chem. Int. Ed.*, **2018**, 57, 9073; Piao, L. et al., *Joule*, **2018**, 2, 549; Chen, X. et al., *Chem. Rev.*, **2019**, 119, 3962; Hyeon, T., *Nat. Mater.*, **2019**, 18, 620; Lou, X. et al., *Adv. Mater.*, **2019**, 31, 1804883; Gong, J. et al., *Adv. Mater.*, **2019**, 31, 1804710; Domen, K. et al. *Nature*, **2020**, 581, 411; *Nat. Commun.*, **2021**, 12: 1005; Wang, X. et al., *Nat. Catal.*, **2020**, 3, 649; Xing, M et al., *Angew. Chem. Int. Ed.*, **2020**, 59, 13968; Negishi, Y. et al., *Angew. Chem. Int. Ed.*, **2020**, 59, 7076; Tanaka, T. et al., *Chem. Sci.*, **2021**, 12, 4940 etc.).

In our previous work, the oxidation cocatalyst is the conventionally used CoO_x cocatalyst that has been widely proved and employed in the photocatalytic system (Zhang, F. et al., *J. Am. Chem. Soc.*, **2012**, 134, 8348; Domen, K. et al., *Energy Environ. Sci.*, **2013**, 6, 3595; Feng, P. et al., *Adv. Mater.*, **2014**, 26, 5043; Li, C. et al., *Angew. Chem. Int. Ed.*, **2015**, 54, 3047; Nathan S. L. et al., *Energy Environ. Sci.*, **2015**, 8, 2644; **2016**, 9, 892; Osamu, I., et al., *J. Am. Chem. Soc.*, **2016**, 138, 14152; Sharp, I. D. et al., *J. Am. Chem. Soc.*, **2017**, 139, 8960). While, in this work, we found that the nanocomposite of CoOOH and FeOOH (denoted as FeCoO_x) produced through the simple *in situ* photodeposition method can act as a more effective oxidation cocatalyst for better performance, Moreover, its structure was analyzed by XAFS measurement in detail to correlate with the performance. Its promotion mechanism on the water oxidation has been well unraveled and discussed as well. After coupling with the Ir cocatalyst to accelerate the reduction of redox, the whole Z-scheme OWS activity is well improved to reach the record value among the redox-driven particular Z-scheme overall water splitting systems. That is to say, the importance and novelty of this work not only develop novel cocatalysts (Ir, FeCoO_x) for promoted water oxidation as well as Z-scheme overall water splitting, but also enhance the scientific understanding to the functionalities of both reduction and oxidation cocatalysts, especially for the FeCoO_x nanocomposite. The fabrication of redox-driven particular Z-scheme overall water splitting system with recorded apparent quantum efficiency (AQE) of 12.3% at 420 nm, and scientific understanding of the structure and working mechanism of cocatalyst

FeCoO_x are believed to be encouraging and interesting for broad readerships working in the field of photocatalysis, especially for water splitting.

As an extended discussion, concerning the judgement of innovation, different people may have different views of standard. However, it should be pointed out that the novelty of one work is normally reflected by one or two areas, and it is impossible for one work to show remarkable novelty in most of areas originating from the novel photocatalyst, novel cocatalyst, surface/interface modification methods/strategies, preparation methodologies of materials or new materials, understanding of science and technology, and promotion of activity etc. Normally, the improvement of activity together with development of strategies, novel materials or improved scientific understanding has been widely published in the journals with extremely high impact like *Science*, *Nature* as well as sister journals. For example, i) BiVO₄ photoanode has been widely investigated (Gamelin, D. R. et al., *J. Am. Chem. Soc.*, **2011**, *133*, 18370; Zou, Z. et al., *Energy Environ. Sci.*, **2011**, *4*, 4046; Choi, K. S. et al., *J. Am. Chem. Soc.*, **2012**, *134*, 2186; *Energy Environ. Sci.*, **2012**, *5*, 8553; *Chem. Soc. Rev.*, **2013**, *42*, 2321; Kro, R. et al., *Nat. Commun.*, **2013**, *4*: 2195; Buddie, C. B. et al., *J. Am. Chem. Soc.*, **2013**, *135*, 11389), but it can be still published in *Science* after the modification of dual cocatalysts FeOOH and NiOOH (Choi, K. S. et al., *Science*, **2014**, *343*, 990). ii) SrTiO₃ has been recently reported to show high AQE under UV irradiation (Domen, K. et al. *Nature*, **2020**, *581*, 411) by integrating previous strategies and preparative methodologies (Domen, K. et al., *J. Catal.*, **2012**, *292*, 26; *J. Mater. Chem. A*, **2016**, *4*, 3027; *Joule*, **2018**, *2*, 509; Li, C. et al., *Energy Environ. Sci.*, **2016**, *9*, 463). iii) C₃N₄ has been widely investigated since it was first reported in 2009 (Wang, X. et al., *Nat. Mater.*, **2009**, *8*, 76; *Angew. Chem. Int. Ed.*, **2012**, *51*, 68; Jaroniec, M. et al., *Adv. Mater.*, **2015**, *27*, 2150; Chai, S. P. et al., *Chem. Rev.*, **2016**, *116*, 7159). However, it has been recently published in *Nature Catalysis* (Wang, X. et al., *Nat. Catal.*, **2020**, *3*, 649) after modulation of crystal facets based on improvement of previous report (Wang, X. et al., *Chem. Sci.*, **2017**, *8*, 550).

Anyway, the development of science and technology is step by step, and no work can completely keep out of previous publications. Judging the novelty of one work

should not be based on the standard whether similar methods and ways have been adopted, but on the standard whether there is new breakthrough in activity, new finding, new material, new strategy or new understanding. Let's encourage each other.

2. FeCoO_x is adopted as the cocatalyst for oxygen evolution in this study. FeCoO_x loaded on BiVO₄ was first reported in Adv. Funct. Mater. 2018, 28, 1802685. Detailed study of the role and impact of FeCoO_x on BiVO₄ for oxygen evolution was reported and supported by theoretical modelling. That pioneering work on FeCoO_x/BiVO₄ is, however, not referred in this manuscript and the reviewer believes it is an important document related to this study.

Response: Thank the reviewer for providing the helpful suggestion. It should be pointed out that in the paper the reviewer mentioned, the FeCoO_x was prepared by photo-assisted electrodeposition method containing first deposition of a thin FeOOH layer and subsequent deposition of CoO_x layer on FeOOH, and finally forming the CoFeO_x through heat treatment. However, the FeCoO_x cocatalyst in this work was simultaneously photodeposited by using the photogenerated holes without any external bias. The deposition method and order are completely different, so their structure should be different as well even though both of them were denoted as FeCoO_x. Moreover, the nanocomposite structure of the FeCoO_x cocatalyst as well as its promotion effect on the water oxidation in this work has been widely characterized and well discussed, while the structure of FeCoO_x in the publication mentioned by the reviewer is unclear. To consider the close correlation of this reference, we have added the paper into the references in the revision.

3. When Ir and FeCoO_x were selectively deposited at the desired sites of BiVO₄, charge transfer boosted by the cocatalyst were maximized. When randomly deposited, the constructive effect was not maximized, but still should be beneficial in general as compared to the bare BiVO₄. The reviewer is curious on why the overall water splitting was not achieved when the cocatalysts were randomly deposited through impregnation. The random location of cocatalyst shouldn't quench totally the Z-scheme reaction.

Response: The random deposition of dual-cocatalysts is truly beneficial for the improvement of performance compared to the bare BiVO₄. As shown in Supplementary Fig. 19, when employing the Ir-CoO_x(Imp.)/BiVO₄ as the OEP, the values of H₂ evolution rate and O₂ evolution rate are both higher than the bare BiVO₄, indicating the random deposition of dual-cocatalysts can promote the O₂-evolving activity but the degree of improvement is relatively limited compared with the activity of selective deposition of dual-cocatalysts. Because of limited improvement, the evolution rate of H₂ is significantly higher than that of O₂ using Ir-CoO_x(Imp.)/BiVO₄ as OEP, leading to the non-stoichiometric of 2:1 for H₂/O₂ and failure to achieve overall water splitting. It should be pointed out that the H₂-evolving rate and O₂-evolving rate should be balanced. Otherwise, the overall water splitting cannot be achieved.

4. When using different HEP (ZrO₂/TaON or MgTa₂O_{6-x}N_y/TaON), similar OWS (H₂ and O₂ evolution) activities were obtained. The authors claimed that it proved the OER on BiVO₄ to be the rate determining. The reviewer agrees that the rate determining step is the OER on BiVO₄. However, OWS through Z-scheme depends on many factors including the interparticulate interaction between OEP and HEP in this work. The similar activities of H₂ and O₂ evolution under different HEP (ZrO₂/TaON or MgTa₂O_{6-x}N_y/TaON) is not clearly understood in this work. If the authors' argument is correct, the same OWS performance will also be observed when the authors fixed the OEP configuration (selective vs random deposition; with single cocatalyst or dual cocatalyst) while switching between the HEP.

Response: To address the reviewer's concern and follow his/her suggestion, we examined the OWS by employing the MgTa₂O_{6-x}N_y/TaON as the H₂-evolving photocatalyst, and employing BiVO₄ with randomly deposited Ir and CoO_x modified or bare BiVO₄ as O₂-evolving photocatalyst. As seen in Fig. R3, their H₂ and O₂ evolution rates are very similar to that using the ZrO₂/TaON as H₂-evolving photocatalyst given in Supplementary Fig. 19. These results further prove that the OER on BiVO₄ is the rate determining.

Fig. R3 Comparison of the Z-scheme OWS performances using BiVO₄ with and without dual-cocatalysts loaded by different methods.

Reaction conditions: 50 mg BiVO₄ with dual-cocatalysts loaded using different methods, 100 mg HEP (MgTa₂O_{6-x}N_y/TaON, 2.5 wt% Rh, 3.75 wt% Cr), 100 mL 25 mM sodium phosphate buffer solution (pH 6.0) containing K₄[Fe(CN)₆] (10 mM), 300 W xenon lamp ($\lambda \geq 420$ nm), temperature: 288 K, Pyrex top-irradiation type.

5. From OCP discussion, FeCoO_x/BiVO₄ has larger band bending than CoO_x/BiVO₄ and therefore better charge transfer. The authors may want to explain further on why such larger band bending was observed. It was suggested in previous study that FeCoO_x forms p-n heterojunction with BiVO₄ and promoted the holes transfer for OER.

Response: Thank the reviewer for providing valuable discussion. To follow the reviewer's suggestion, the following sentence has been added into the revision for extended discussion (Highlighted in yellow background in Page 11 in the revision).

“the more intense band bending should result from the p-n heterojunction between FeCoO_x and BiVO₄⁴¹.”

Reviewer #3:

The authors demonstrate the enhancement on the water oxidation performance of BiVO_4 using novel cocatalysts that have been prepared via in-situ photodeposition. The author also performed various and complementary analyses including DFT and XAS to find out the mechanism of water oxidation promotion in their system. This manuscript provides creditable results and reasonable discussion which is related to the results. I would recommend the publication of this manuscript after addressing the following comments.

1) The authors claimed that the OWS system in this study shows good photostability. However, the stability test time in Supplementary Figure 19 is not enough to support this claim. The authors should provide the stability test for at least more than 50 hours under the illumination of both 300 W xenon lamp ($\lambda \geq 420 \text{ nm}$) and solar simulator (1 sun).

Response: To follow reviewer's suggestion, we examined multiple cycles of curve of Z-scheme OWS under AM 1.5G. As shown in Fig. R4 and in Supplementary Fig. 21, the steady increase of H_2 and O_2 evolution can be observed, demonstrating good photostability of the system. To make it clear, the following description has been added into the revision (Highlighted in yellow background in Page 15 in the revision).

“The multiple cycles of time-course curves shown in Supplementary Fig. 20 and 21 demonstrate the good photostability of the system constructed in this study.”

Fig. R4 Multiple cycles of curve of Z-scheme OWS under illumination of the standard solar simulator (AM 1.5G, 100 mW cm⁻²).

Reaction conditions: 50 mg OEP (0.8 wt% Ir; 0.2 wt% Co), 50 mg HEP (ZrO₂/TaON, 1.0 wt% Rh, 1.5 wt% Cr), 100 mL 25 mM sodium phosphate buffer solution (pH 6.0) containing K₄[Fe(CN)₆] (10 mM), temperature: 288 K, Pyrex top-irradiation type.

2) *The authors showed only the UV-Vis spectra of Ir-FeCoO_x/BiVO₄ OEP. I'm wondering if there is any difference in the light absorption property of bare BiVO₄ and Ir-FeCoO_x/BiVO₄. The authors should provide the UV-Vis spectra of both OEPs.*

Response: To follow the reviewer's suggestion, we test the UV-Vis spectra of bare BiVO₄ and Ir-FeCoO_x/BiVO₄, and the result is given in Fig. R5 and in Supplementary Fig. 2. It is shown that after loading the dual-cocatalysts, the absorption of long wavelength range is obviously increased, indicating the successful deposition of dual-cocatalysts. Additionally, the absorption edge of the BiVO₄ is not obviously changed.

Fig. R5 UV-Vis DRS of the BiVO₄ and Ir-FeCoO_x/BiVO₄.

In order to make it clear, the following description has been added into the revision (Highlighted in yellow background in Page 6 in the revision).

“And the change in the long wavelength range of UV-Vis diffuse reflectance spectra (DRS) can confirm the successful deposition of the dual-cocatalysts (Supplementary Fig. 2).”

REVIEWER COMMENTS

Reviewer #1 (Remarks to the Author):

Reviewer #1 Report on “Unraveling of cocatalysts...” by Y. Qi et al (Nature Comm., Ref. NCOMMS-21-16144A)

I have gone through the revised material carefully including the 18-page response/rebuttal document. I appreciate the fact that the authors have done a thorough job of addressing all the points raised during the initial submission of this manuscript.

I was VERY ANNOYED by the authors’ rather condescending remarks and LECTURE about the “basic concepts in the field of photocatalysis” (p. 7). This is NOT what I was looking for after spending some 4 decades in this field!! Then the authors proceed to spend the next page and a half with hand-waving statements and superfluous references without getting to the point!

a) I still maintain that the parent BiVO₄ surface is NOT catalytically active (see the authors’ own Figures 3a and c). Thus the use of the word “co” as in “cocatalyst” has no meaning whatsoever! I would insist that the authors avoid this WRONG terminology including in the title and replace “cocatalyst” everywhere in the text with simply “catalyst”.

The fact that 100 other authors (I am being facetious here!) have used the term: ‘cocatalyst’ does not necessarily mean that this terminology is correct; Pied Piper comes to mind here.

b) I am NOT going to insist, however, that the authors change their (wrong, at least in my reckoning) use of the word: “photocatalysis” or “photocatalyst” since some argument can be made for retention of these words’ use.

Reviewer #2 (Remarks to the Author):

The authors recently reported CoOx and Au on BiVO₄ for Z-scheme overall water splitting. In this report, CoOx is replaced by FeCoOx, while Au is replaced by Ir. Activity is improved, working mechanism is proposed. The authors elaborated the importance of cocatalyst, in which the reviewer totally agree. However, introducing FeOOH to CoOOH to form FeCoOx and replacing Au with another expensive metal Ir, did not provide sufficient novelty and scientific advancement for paper to be accepted by Nature Communications. Furthermore, FeCoOx as cocatalyst for oxygen evolution has been studied thoroughly (experimentally and theoretically) in Adv Funct Mater. In this regards, the novelty of FeCoOx as new cocatalyst in this work is also a weakness.

Reviewer #3 (Remarks to the Author):

The authors have addressed and revised the Reviewers' comments carefully and correctly. Therefore, I would recommend the publication of this paper in Nature Communications.

Reviewer #4 (Remarks to the Author):

The authors have discussed the influence of cocatalysts Ir and FeCoOx on the surfaces of BiVO4. They have shown experimentally that the presence of the cocatalysts enhanced the OER, particularly with the dual deposition Ir-FeCoOx. Dual cocatalysts deposition is a practice considered for many other OER efficient materials, and particularly for BiVO4 [RSC Adv., 2017,7, 15053-15059]. To 'unravel' (as stated by the authors) the mechanism for water oxidation with the role of both cocatalysts together with the effect on surface selectivity can bring relevant information to guarantee a publication on Nature Communications. However such mechanisms are not well supported, particularly on the theory side.

1) On the main text the author state that the in gap surface band appearing on FeCoOx/BiVO4 is mainly composed of Co 3d and O 2p states while Fe states show slight contribution. However, from the DOS presented, Fe atoms contribute as much as the O 2p for the particular band.

How do such surface states are concerning the Ox-potential, that is, how the flat band potential is changing with the surface modification. Part of the experimental conclusion elucidates a better charge separation and electron transfer of FeCoOx compared to CoOx, and this information could be relevant for the microscopic origin of such enhancement.

Additionally, the discussion was rather cumbersome to read. Half of the theoretical discussion refers to figures on the SM. The authors could bring the Figure 14 of the SM to the main text Figure 4, putting the DOS below the respective slab associated with it.

2) A important discussion missing is the validity of the proposed slab model concerning the experimental sample. The proposed FeCoOx and CoOx surface structures are comparable with the bond lengths observed on the experimental radial spectra? But more important, is the charge state of Co and Fe

comparable with the experimentally observed? This information can be extracted from the XANES, or even compared with XANES simulations.

Additionally, the finite slab with only 7 Angstrom thickness is enough to represent the surface of BiVO₄? Given the asymmetry of such slab in the presence of CoOx and FeCoOx, the charge transfer between the upper and lower surfaces can be influential on the active sites' charge states, not corresponding to the experimental scenario. The authors should validate their conclusions in this regard.

3) The role of the Ir and also the surface selectivity, which could be enlightened by the DFT calculations, were not considered here.

Responses to the reviewer(s)' comments

First of all, we would like to thank all the referees for spending their valuable time on reviewing our manuscript. In the first revision, we have carefully addressed the concerns raised by the three experimental experts, and are now glad to see that all of them have given kindly positive comments and recommendation except for minor questions/concerns that will be concisely replied in consideration of the fact that we have given a detailed discussion and explanation on them in the first revision. In this revision, we will mainly address the concerns/comments raised by the new referee who is focused on the theoretical calculation. Please see the details point-by-point as below.

Reviewer #1:

I have gone through the revised material carefully including the 18-page response/rebuttal document. I appreciate the fact that the authors have done a thorough job of addressing all the points raised during the initial submission of this manuscript. I was VERY ANNOYED by the authors' rather condescending remarks and LECTURE about the "basic concepts in the field of photocatalysis" (p. 7). This is NOT what I was looking for after spending some 4 decades in this field!! Then the authors proceed to spend the next page and a half with hand-waving statements and superfluous references without getting to the point!

a) I still maintain that the parent BiVO_4 surface is NOT catalytically active (see the authors' own Figures 3a and c). Thus the use of the word "co" as in "cocatalyst" has no meaning whatsoever! I would insist that the authors avoid this WRONG terminology including in the title and replace "cocatalyst" everywhere in the text with simply "catalyst".

The fact that 100 other authors (I am being facetious here!) have used the term: 'cocatalyst' does not necessarily mean that this terminology is correct; Pied Piper comes to mind here.

b) I am NOT going to insist, however, that the authors change their (wrong, at least in my reckoning) use of the word: "photocatalysis" or "photocatalyst" since some

argument can be made for retention of these words' use.

Response: Thank the reviewer for spending his/her valuable time on reviewing our revision again and giving an integral positive remark on it. We really admire the reviewer's professional spirit on science like the definition of academic terminology. Concerning the usage of cocatalyst, we would like to give further detailed response that the parent BiVO₄ surface is catalytically active for both water oxidation and reduction of the [Fe(CN)₆]³⁻ ions used in our system based on the following facts: i) as seen in Fig. 3a, remarkable electrochemical cathode current (black line) can be observed for the parent BiVO₄. This well reveals that the [Fe(CN)₆]³⁻ ions can be reduced by BiVO₄ itself. That is to say, the BiVO₄ is catalytically active for reduction of [Fe(CN)₆]³⁻ ions. ii) as shown in Fig. 3c, obvious photoanode current can be observed for the pristine BiVO₄ (green line). This indicates that the surface of BiVO₄ is catalytically active for water oxidation. iii) as depicted in Supplementary Fig. 18, when the content of Ir loaded is zero, the parent BiVO₄ can remarkably drive the photocatalytic water oxidation in the presence of [Fe(CN)₆]³⁻ ions, whose O₂ evolution rate is ca. 3 μmol/h. This further reveals that the BiVO₄ itself is catalytically active for both reduction of [Fe(CN)₆]³⁻ ions and water oxidation. Additionally, it is worth noting that the water oxidation can be well promoted after the loading of FeCoO_x regardless of particulate photocatalysis (Supplementary Fig. 19) or photoanode (Fig. 3c), and the reduction of [Fe(CN)₆]³⁻ ions can be promoted by the deposition of Ir (Fig. 3a). To follow above facts, we consider it reasonable to call the deposited Ir and FeCoO_x as cocatalyst. Surely, we can understand that the main catalytic center has been widely called as catalyst in the field of electrocatalysis or thermal catalysis. However, if the parts of photocatalytic system is separately called as “photocatalyst and catalyst” instead of “photocatalyst and cocatalyst”, it will be confusing as well. We would like to follow the habit in the field of photocatalysis. Thank the reviewer for useful discussion very much.

Reviewer #2:

The authors recently reported CoO_x and Au on BiVO₄ for Z-scheme overall water splitting. In this report, CoO_x is replaced by FeCoO_x, while Au is replaced by Ir. Activity

is improved, working mechanism is proposed. The authors elaborated the importance of cocatalyst, in which the reviewer totally agree. However, introducing FeOOH to CoOOH to form FeCoO_x and replacing Au with another expensive metal Ir, did not provide sufficient novelty and scientific advancement for paper to be accepted by Nature Communications. Furthermore, FeCoO_x as cocatalyst for oxygen evolution has been studied thoroughly (experimentally and theoretically) in Adv Funct Mater. In this regards, the novelty of FeCoO_x as new cocatalyst in this work is also a weakness.

Response: Thank the reviewer for spending his/her valuable time on reviewing our revision. We are glad to see that the reviewer agree with the importance of developing cocatalyst. However, it seems that he/she is still misunderstanding to the difference between the FeCoO_x cocatalyst denoted in this work and that in previous publication (*Adv. Funct. Mater.* the reviewer mentioned). Here we would like to emphasize once more that the structure and preparative methodologies of FeCoO_x cocatalyst in our work is completely different from that in the *Adv. Funct. Mater.*, even though both of them are similarly denoted as FeCoO_x. First of all, the FeCoO_x cocatalyst in this work was produced through the simultaneous *in situ* photo-deposition method by using the photogenerated holes free of any external bias, while the CoFeO_x in the mentioned publication was step-by-step prepared by photo-assisted electrodeposition method containing first deposition of a thin FeOOH layer, subsequent deposition of CoO_x layer and final calcination. Since the deposition method and order are completely different, their local structure and composition should be largely distinct. Secondly, the structure of FeCoO_x in this work has been well disclosed to exist as nanocomposite of CoOOH and FeOOH (denoted as FeCoO_x for simplicity), while the local structure of the FeCoO_x in the previous AFM publication is unclear. Thirdly, the underlying mechanism about the promotion effect of the FeCoO_x nanocomposites on the water oxidation has been well discussed in this work. We do hope that now it is clear for the reviewer.

Concerning the novelty of this work, we would like to highlight the following points again: i) Both Ir and FeCoO_x nanocomposites have been developed as new efficient cocatalysts. ii) The local structure and underlying working mechanism of the

cocatalysts developed here (especially for the FeCoO_x) have been unveiled to get deep scientific insight. iii) Based on the breakthrough of above two areas, we finally constructed the highly efficient Z-scheme overall water splitting system with a benchmarked apparent quantum efficiency of 12.3% at 420 nm.

Reviewer #3: *The authors have addressed and revised the Reviewers' comments carefully and correctly. Therefore, I would recommend the publication of this paper in Nature Communications.*

Response: Thank the reviewer for his/her kind recommendation very much.

Reviewer #4:

The authors have discussed the influence of cocatalysts Ir and FeCoO_x on the surfaces of BiVO_4 . They have shown experimentally that the presence of the cocatalysts enhanced the OER, particularly with the dual deposition Ir- FeCoO_x . Dual cocatalysts deposition is a practice considered for many other OER efficient materials, and particularly for BiVO_4 [RSC Adv., 2017,7, 15053-15059]. To 'unravel' (as stated by the authors) the mechanism for water oxidation with the role of both cocatalysts together with the effect on surface selectivity can bring relevant information to guarantee a publication on Nature Communications. However such mechanisms are not well supported, particularly on the theory side.

1) On the main text the author state that the in gap surface band appearing on $\text{FeCoO}_x/\text{BiVO}_4$ is mainly composed of Co 3d and O 2p states while Fe states show slight contribution. However, from the DOS presented, Fe atoms contribute as much as the O 2p for the particular band. How do such surface states are concerning the Ox-potential, that is, how the flat band potential is changing with the surface modification. Part of the experimental conclusion elucidates a better charge separation and electron transfer of FeCoO_x compared to CoO_x , and this information could be relevant for the microscopic origin of such enhancement. Additionally, the discussion was rather cumbersome to read. Half of the theoretical discussion refers to figures on the SM. The

authors could bring the Figure 14 of the SM to the main text Figure 4, putting the DOS below the respective slab associated with it.

Response: Thank the new reviewer for spending his/her valuable time on reviewing our revision, especially for the theoretical part. It is really useful discussion and comment and very helpful for us to improve the quality of this work. We will address and give the responses point-by-point as follows.

i) Concerning the underestimated contribution of the Fe 3d state in our previous description, we have corrected it by revising our description with following sentence added in the revised manuscript (Highlighted in yellow background in Page 13 in the revision).

“a mixed band mainly composed of Co 3d, Fe 3d and O 2p states emerges between the valence band and conduction band (Fig. 4g).”

Additionally, to follow the reviewer’s suggestion, we have moved the original Supplementary Fig. 14 into Figure 4f-h in the revision.

ii) As for the flat band potential change after the surface modification, we tested the flat-band potentials (V_{fb}) of the BiVO_4 , $\text{CoO}_x/\text{BiVO}_4$ and $\text{FeCoO}_x/\text{BiVO}_4$ through Mott-Schottky. As shown in Figure R1, after loading the cocatalyst, the V_{fb} shows cathodic shift and the FeCoO_x displays the greatest extent, which indicates the surface states caused by the partial Fermi level pinning maybe be suppressed. By comparing with the turn on voltage of different photoanodes in Figure 3c, the shift trend is well consistent, implying the FeCoO_x cocatalyst has a better suppression effect on the surface state. It should be pointed out that the further discussion about the surface states is interesting but beyond the scope of this work.

Figure R1. The Mott-Schottky curves of BiVO_4 , $\text{CoO}_x/\text{BiVO}_4$ and $\text{FeCoO}_x/\text{BiVO}_4$.

iii) To prove the better charge separation and electron transfer of FeCoO_x compared to CoO_x from the theoretical calculation, we compare the valence band of BiVO_4 , CoO_x and FeCoO_x , and calculate the d-band center (E_d) of active sites. As seen in Figure R2, the valence band of BiVO_4 and CoO_x in $\text{CoO}_x/\text{BiVO}_4$ is almost the same, but the valence band of FeCoO_x is about 0.8 eV higher than that of BiVO_4 in $\text{FeCoO}_x/\text{BiVO}_4$. Based on them, we can deduce that the bigger valence band difference between BiVO_4 and FeCoO_x indicates the larger driving force for charge separation compared with CoO_x . What's more, it has been demonstrated by the d-band center theory^[1,2] that if the d-band center of the active sites for the same element is closer to the Fermi surface, the metal activity will become higher and the adsorption will become stronger. As given in Figure R3, the E_d value of Co in the $\text{FeCoO}_x/\text{BiVO}_4$ catalyst was calculated as -1.63 eV, sharply increased compared to that (-2.56 eV) of Co in the $\text{CoO}_x/\text{BiVO}_4$ catalyst. This means that the introduction of Fe has effectively modulated the electronic structure of Co on the $\text{FeCoO}_x/\text{BiVO}_4$ to own much stronger adsorption properties for the OER intermediates with respect to the $\text{CoO}_x/\text{BiVO}_4$, leading to smoother charge transfer from FeCoO_x to water for production of oxygen.

Figure R2. The DOS of BiVO₄ and CoO_x in CoO_x/BiVO₄ (a), BiVO₄ and FeCoO_x in FeCoO_x/BiVO₄ (b).

Figure R3. Comparison of d-band center of active Co site on FeCoO_x/BiVO₄ and CoO_x/BiVO₄.

To address the reviewer's concerns, the following sentences have been added into the revision for extended discussion (Highlighted in yellow background in Page 14 in the revision).

“Additionally, as shown in Supplementary Fig.15, the d-band center (Ed) value of Co active sites in FeCoO_x/BiVO₄ was calculated as -1.63 eV, which are sharply increased with respect to the CoO_x/BiVO₄ (-2.56 eV). This demonstrates that the electronic structure of Co active sites can be well modulated and optimized in the FeCoO_x/BiVO₄ due to the introduction of Fe atoms to get much stronger adsorption properties to the OER intermediates according to the d-band center theory^{44,45}.”

2) A important discussion missing is the validity of the proposed slab model concerning the experimental sample. The proposed FeCoO_x and CoO_x surface structures are comparable with the bond lengths observed on the experimental radial spectra? But more important, is the charge state of Co and Fe comparable with the experimentally observed? This information can be extracted from the XANES, or even compared with XANES simulations. Additionally, the finite slab with only 7 Angstrom thickness is enough to represent the surface of BiVO₄? Given the asymmetry of such slab in the presence of CoO_x and FeCoO_x, the charge transfer between the upper and lower surfaces can be influential on the active sites' charge states, not corresponding to the experimental scenario. The authors should validate their conclusions in this regard.

Response: Thank the new reviewer for his/her interesting discussion and comments. Here we want to emphasize that the model built is based on the results of XANES measurement. In the Table S1, the detailed structure parameters including coordination numbers and bond lengths were given after the reasonable fitting.

Concerning the charge state of Co and Fe, the valence state change of Co active site with and without Fe introduction was evaluated by the bader charge method. As seen in Table R1, the valence state of Co in the FeCoO_x/BiVO₄ sample is higher than that in the CoO_x/BiVO₄, as is in accordance with the trend observed in the Co K-edge XANES tests (Figure R4). This means that the introduction of Fe will enhance the valence state of Co and the Co active site in FeCoO_x/BiVO₄ possesses stronger oxidation capacity, which is beneficial for OER reaction.

Table R1. Calculated Bader charge for Fe and Co within the CoO_x/BiVO₄ and

FeCoO_x/BiVO₄.

	Co in CoO _x /BiVO ₄	Co in FeCoO _x /BiVO ₄	Fe in FeCoO _x /BiVO ₄
Bader charge	1.2 a.u.	1.3 a.u.	1.6 a.u.

Figure R4. The Co K-edge XANES $\mu(E)$ spectra of the CoO_x/BiVO₄ and FeCoO_x/BiVO₄.

It should be pointed out that although Mulliken charge and Bader charge have been employed to calculate the valence electron configuration in some solid systems, it is still difficult to get the exact charge distribution according to our experience and previous calculation examples^[3,4], especially when there is a vacuum layer in calculation. Additionally, valence electron configurations with the Hybrid functional correction will become more complex and difficult to analyze.

To address the reviewer's concerns, the following sentences have been added into the revision for extended discussion (Highlighted in yellow background in Page 13 and Page 14 in the revision).

“In order to microscopically understand the better electron transfer on the FeCoO_x with respect to the CoO_x, their Bader charges were calculated and compared. As given in Supplementary Table 4, the changing trend of Bader charge on the Co active site after introduction of Fe (increase from 1.2 a.u. in CoO_x/BiVO₄ to 1.3 a.u. in FeCoO_x/BiVO₄) is in line with the changing one of experimental valence state (Supplementary Fig. 14).

Compared to the $\text{CoO}_x/\text{BiVO}_4$, the higher bader charge on the Co active site in $\text{FeCoO}_x/\text{BiVO}_4$ indicates its stronger oxidation capacity as well as more beneficial electron transfer⁴³.”

As for the reasonability and validity of the thickness, we think it reasonable and valid because of the following facts: i) in our BiVO_4 slab, 7 Angstrom thickness is composed of around 5 to 6 Vanadium or Bismuth oxide layers. These number of oxide layers on substrate is commonly used in many previous OER literature^[5-9]. ii) the charge transfer in $\text{FeCoO}_x/\text{BiVO}_4$ or $\text{CoO}_x/\text{BiVO}_4$ between the cations is generally through the bridging oxygen molecule. Generally speaking, the interaction between non-adjacent cations is very weak. As far as $\text{FeCoO}_x/\text{BiVO}_4$ is concerned, only the Co-O-Fe or Co-O-Bi on the outside surface should own the governing charge transfer to the active Co site. iii) to our knowledge, the interaction that can penetrate 5 more Angstroms is the long-range weak. It is commonly seen in molecular crystals and hydrogen bond dominated systems^[10], which is not suitable for this article, either.

We do agree with the reviewer that the thicker model is better. However, it is worthy of being pointed out that the hybrid functional PBE0 used in this work to calculate the precise electronic structure and band gap, consumes about 100 times of memory compared to the ordinary PBE functional, and the current calculation based on about 100 atoms of thickness in this work has reached the memory limit of the current DFT calculation.

3) The role of the Ir and also the surface selectivity, which could be enlightened by the DFT calculations, were not considered here.

Response: Thanks for interesting discussion. It should be pointed out that the Ir nanoparticles are selectively photo-deposited on the {010} facet of BiVO_4 (Figure 2b) to collect the photo-generated electrons and promote the reduction of the redox $[\text{Fe}(\text{CN})_6]^{3-}$ ions instead of promoting OER process. Concerning the function and role of deposited Ir, please refer to the illustration on the OEP in Figure 1 and the LSV result given in Figure 3a with simple discussion already given in the first paragraph of page 10. For simplicity, the role of deposited Ir is to promote electron transfer from

conduction band of BiVO₄ to its surface and to accelerate the reduction kinetics of [Fe(CN)₆]³⁻ ions acting as cocatalyst. Concerning the surface selectivity, it has been discussed in detail in our previous publication (Nat. Commun. 2013, **4**, 1432, cited as reference 32 in this work). It was found that the noble metal (such as Au, Ag, Pt) was selectively deposited on the {010} facet of BiVO₄ by the photo-generated electrons and the metal oxide (such as PbO₂, MnO_x) was selectively deposited on the {110} facet of BiVO₄ by the photo-generated holes. That is to say, the photo-generated electrons and holes transfer to the different facets to participate in the different reactions caused by the spatial charge separation. The selective photodeposition of Ir on the {010} facets has been confirmed by the Figure 2a and b in this work.

References:

- [1] Norskov J K, Chemisorption on Metal-Surfaces. Reports on Progress in Physics, 1990, 53(10): 1253-1295.
- [2] Norskov J K, Electronic Factors in Catalysis. Progress in Surface Science, 1991, 38(2): 103-144.
- [3] Zhang B, Wang L, Cao Z, et al. High-valence metals improve oxygen evolution reaction performance by modulating 3d metal oxidation cycle energetics. Nature Catalysis, 2020, 3(12): 985-992.
- [4] Zhou D, Cai Z, Bi Y, et al. Effects of redox-active interlayer anions on the oxygen evolution reactivity of NiFe-layered double hydroxide nanosheets. Nano Research, 2018, 11(3) 1358-1368.
- [5] Man I C, Su H Y, Calle - Vallejo F, et al. Universality in oxygen evolution electrocatalysis on oxide surfaces[J]. ChemCatChem, 2011, 3(7): 1159-1165.
- [6] Grimaud A, Diaz-Morales O, Han B, et al. Activating lattice oxygen redox reactions in metal oxides to catalyse oxygen evolution[J]. Nature chemistry, 2017, 9(5): 457-465.
- [7] Ng J W D, García-Melchor M, Bajdich M, et al. Gold-supported cerium-doped NiO_x catalysts for water oxidation[J]. Nature Energy, 2016, 1(5): 1-8.
- [8] Grimaud A, May K J, Carlton C E, et al. Double perovskites as a family of highly active catalysts for oxygen evolution in alkaline solution[J]. Nature communications, 2013, 4(1): 1-7.
- [9] Zhang J, Wang T, Pohl D, et al. Interface engineering of MoS₂/Ni₃S₂ heterostructures for highly enhanced electrochemical overall - water - splitting activity[J]. Angewandte Chemie, 2016, 128(23): 6814-6819.
- [10] Kong Y, Hou D, Zhang H D, et al. Davydov Collective Vibrational Modes and Infrared Spectrum Features in Aniline Crystal: Influence of Geometry Change Induced by van der Waals Interactions[J]. The Journal of Physical Chemistry C, 2017, 121(34): 18867-18875.

REVIEWERS' COMMENTS

Reviewer #4 (Remarks to the Author):

The authors have addressed all my concerns, providing further evidence on the theory side. I believe that the present version of the manuscript is suited for publication on Nature Communications.

Responses to the reviewer(s)' comments

We would like to thank the referee for spending their valuable time on reviewing our revised manuscript. To well address the reviewers' concern, we will give a detailed response point by point as below.

Reviewer #4:

The authors have addressed all my concerns, providing further evidence on the theory side. I believe that the present version of the manuscript is suited for publication on Nature Communications.

Response: Thank the reviewer for his/her kind recommendation very much.